# Metamorphic Testing of Machine Learning and Conceptual Hydrologic Models

Peter Reichert[1,*], Kai Ma[2,3], Marvin Höge[1], Fabrizio Fenicia[1], Marco Baity-Jesi[1], Dapeng Feng[4], and Chaopeng Shen[4]

[1]Eawag: Swiss Federal Institute of Aquatic Science and Technology, Dübendorf, Switzerland
[*]current status: retired from Eawag, see https://peterreichert.github.io for updated information
[2]Institute of International Rivers and Eco-Security, Yunnan University, Kunming, China
[3]Yunnan Key Laboratory of International Rivers and Transboundary Eco-security, Yunnan University, Kunming, China
[4]Civil and Environmental Engineering, Pennsylvania State University, University Park, PA, USA

**Correspondence:** peter.reichert@emeriti.eawag.ch; https://peterreichert.github.io

**Abstract.** Predicting the response of hydrologic systems to modified driving forces, beyond patterns that have occurred in the past, is of high importance for estimating climate change impacts or the effect of management measures. This kind of predictions requires a model, but the impossibility of testing such predictions against observed data makes it difficult to estimate their reliability. Metamorphic testing offers a methodology for assessing models beyond validation with real data. It consists of defining input changes for which the expected responses are assumed to be known at least qualitatively, and to test model behavior for consistency with these expectations. To increase the gain of information and reduce the subjectivity of this approach, we extend this methodology to a multi-model approach and include a sensitivity analysis of the predictions to training or calibration options. This allows us to quantitatively analyse differences in predictions between different model structures and calibration options in addition to the qualitative test to the expectations. In our case study, we apply this approach to selected conceptual and machine learning hydrological models calibrated to basins from the CAMELS data set. Our results confirm the superiority of the machine learning models over the conceptual hydrologic models regarding the quality of fit during calibration and validation periods. However, we also find that the response of machine learning models to modified inputs can deviate from the expectations and the magnitude and even the sign of the response can depend on the training data. In addition, even in cases in which all models passed the metamorphic test, there are cases in which the quantitative response is different for different model structures. This demonstrates the importance of this kind of testing beyond and in addition to the usual calibration-validation analysis to identify potential problems and stimulate the development of improved models.

# 1 Introduction

The availability of hydrologic and meteorological data and catchment attributes for a large number of catchments in the USA (Newman et al., 2015; Addor et al., 2017) has greatly stimulated hydrologic research in the past few years (Kratzert et al., 2018; Shen, 2018; Kratzert et al., 2019a, b; Razavi, 2021; Ng et al., 2023; Feng et al., 2020). In particular, it has been shown that the training of machine learning models jointly to hydrologic data from a large number of catchments leads to an extraordinary performance of these models even for the prediction of the output of catchments that had not been used for training (Kratzert et al., 2018, 2019a, b; Feng et al., 2020, 2021). Arguably, this breakthrough was made possible by the combination of two elements:

  (i) using machine learning models, in particular deep learning architectures in the form of Long Short-Term Memory (LSTM) models, that are highly flexible and contain a large number of parameters;

  (ii) training the models jointly on large sets of diverse catchments using relevant catchment attributes as additional input to meteorological time series to allow the models to learn diverse response patterns and their dependence on catchment characteristics.

Due to the use of a large and diverse data set, overfitting of the models is mitigated and the models to some degree gain the capability of acquiring hydrologic knowledge (Kratzert et al., 2018, 2019a, b). It has been shown that this kind of hydrologic knowledge can even be transferred across continents (Ma et al., 2021). The success demonstrated by a large number of studies based on machine learning models trained on such data sets has challenged the belief of hydrologists that the prediction of the output of ungauged catchments would only be possible with models that are built with strong support by hydrologic expert knowledge (Hrachowitz et al., 2013; Nearing et al., 2021). The availability of many more data sets for other countries than the USA, such as Chile (Alvarez-Garreton et al., 2018), Great Britain (Coxon et al., 2020), Brazil (Chagas et al., 2020), Australia (Fowler et al., 2021), Switzerland (Höge et al., 2023), and more, bears great potential for further development of hydrologic modelling across catchments, continents, and climatic regions.

The primary focus of the studies cited above was on model training and validation on a future part of the time series or on catchments not used for calibration. The question whether this success is transferable to the prediction of the consequences of modified driving forces in these catchments has been less investigated (Bai et al., 2021; Natel de Moura et al., 2022; Wi and Steinschneider, 2022). For the prediction of the effects of climate change and water management measures on the hydrology of catchments, it is of particular interest to modify driving forces beyond the patterns observed in the past. When the perturbation is large enough, there is no data available for validating the models under such perturbations. The problem is of different nature for conceptual hydrologic models than for machine learning models. The prediction of the behavior of catchments under modified driving forces with conceptual models is challenging because it is very hard to predict the required modifications to model parameters induced by changes in vegetation, soil structure, etc. (Merz et al., 2011). And it is also difficult to extend the models to mechanistically describe these changes. The prediction with machine learning models could either lead to wrong

results due to poor out-of-domain generalization (Wang et al., 2022), or the results could be much better due to a more comprehensive consideration of adapted catchment properties learned from other catchments in the training set. Which of these effects dominates may depend on the degree of input modifications and on the diversity of the set of catchments used for training. For these reasons, it is of interest to compare predictions of both kinds of models under modified driving forces and to investigate whether the results depend on the training data set and on parameters of the optimization algorithm.


It is the goal of this study to compare the behavior of machine-learning and conceptual models under modified driving forces and to investigate to which degree we can learn about deficiencies of models and pathways for their improvement from these results. Such attempts have been done before and have uncovered problems in the predictions of LSTM models (Bai et al., 2021; Razavi, 2021; Natel de Moura et al., 2022; Wi and Steinschneider, 2022). We extend this kind of studies by 60 considering precipitation changes in addition to temperature changes (this has been done in some of the previous studies), by using LSTM models trained on a large set of catchments (this has been done in some of the previous studies), by investigating responses for different elevation classes separately to reduce the uncertainty in the response predicted by the experts, and by including sensitivity analysis regarding catchment attributes, basins used for calibration, and numerical seed of the optimization algorithm. We will do model simulations with isolated changes in precipitation and temperature and compare the resulting 65 change in outlet discharge with the expected outcomes for selected basins from the CAMELS data set (Newman et al., 2015; Addor et al., 2017). Note that this is a metamorphic testing design (Xie et al., 2011; Yang and Chui, 2021) that facilitates the formulation of the qualitative expected behavior, rather than a realistic climate change scenario that would consist of coupled temperature and precipitation changes with more complex time dependence. Based on this design, the more specific goals of our study are to answer the following questions:

1. Are good fits during calibration and validation periods sufficient to gain confidence in predictions under modified driving forces?

    2. How useful is metamorphic testing of models beyond the usual calibration-validation analysis?

    3. Do machine learning models always improve when extending the training data set?

    4. How do machine learning and conceptual models complement each other in terms of strengths and deficits?

## 2   Methods

### 2.1   Metamorphic Testing

Metamorphic testing is a methodology for assessing models beyond validation with real data (Xie et al., 2011; Yang and Chui, 2021). It consists of

    (i) defining changes to model input for which the expected response of the underlying system is assumed to be known at 80        least qualitatively, and

(ii) testing the model response to these changes for consistency with these expectations.

Note that metamorphic testing does not replace calibration and validation but it is an additional, complementary test to the quality of fit that is specifically targeted at situations (inputs) for which there are no response data available. The input changes underlying metamorphic testing should be designed in such a way that they reflect aspects of inputs that are of interest for predicted output but that they still allow for a qualitative characterization of expected responses. One methodology to design such input changes is to reduce the dimension of the problem by modifying just one input with a relatively simple pattern rather than using correlated input changes in multiple inputs and complicated temporal pattern as it would be needed for real predictions. Such additional tests to model fit are important as it has been shown that quality of fit and prediction accuracy do not necessarily improve in parallel. At least one case study came to the conclusion: "Surprisingly, the prediction accuracy of a model and its ability to provide consistent predictions were found to be uncorrelated" (Yang and Chui, 2021). The conclusions may not always be that extreme, but such cases indicate the need for model testing beyond the quality of fit.

The weakness of metamorphic testing is that it requires the specification of the expected response of a system under modified inputs. Even if we define simple input changes to facilitate the fulfillment of this requirement, it still requires partly subjective expert judgements that may be biased by the limited mechanistic understanding of the system's function by the experts or, more generally, by the incomplete state of current scientific knowledge. To further increase the understanding of model behavior and reduce the subjectivity of testing, we use a multi-model approach and extend the test to the analysis of the sensitivity of the results to model structure and to different training or calibration options. In particular, we compare conceptual and machine learning approaches as we expect complementary strengths and weaknesses. Conceptual approaches, due to their consideration of (simplified) physical principles, can be expected to provide reliable predictions if the input changes are small enough to not considerably alter catchment properties, such as vegetation and soil structure. On the other hand, machine learning models may be more critical for out-of-sample predictions, but due to a high diversity of catchments used for training, they bear the potential of considering also changes in catchment properties. This allows us to identify quantitative deviations of predictions (to modified inputs) between model structures. The investigation of the sensitivity of the predictions to calibration options further provides insight into the robustness of the results of the metamorphic test. The chosen model structures are described in more detail in section 2.2 and in the appendices A and B, the complementary calibration option in section 3.3.

There are four potential outcomes of this extended metamorphic testing approach:

**(A)** **Metamorphic test succeeded, models mutually consistent.** The predicted response is robust against the investigated model structures and changes in the calibration process and agrees with the expectations. This result confirms the model structures and increases the trust into reliable predictions.

**(B)** **Metamorphic test succeeded, but quantitative responses of different models disagree.** The predicted response is in qualitative agreement with the expectations, but the quantitative response is sensitive to the investigated model structures

or to aspects of the model calibration process. This result shows the limits of metamorphic testing but the identified differences between responses may still stimulate thinking about model structure improvements.

**(C)** **Metamorphic test failed, some models inconsistent with others.** The predicted response is sensitive to the investigated model structures or aspects of the model calibration process with some responses in agreement and others in disagreement with the expert expectations. This indicates problems of some models to reliably predict the response to the investigated input changes and indicates the need for a revision of model structures or training processes.

**(D)** **Metamorphic test failed, models mutually consistent.** The predicted response is robust against the investigated model structures and changes in the calibration process but it disagrees with the expectations. This clearly demonstrates a serious problem either caused by similar deficits of all model structures that lead to wrong predictions or to incomplete scientific knowledge that lead to incorrect expert predictions. This is the most difficult outcome of the metamorphic analysis but it still demonstrates its importance as it uncoveres a problem. In this case it is very important to think of potential mechanisms that may have been overlooked by the experts as well as similar structural deficits of all investigated models. This may initiate an extended research process that depends on the investigated system and models.

For metamorphic testing we choose simple, isolated changes in precipitation (increase by 10%) and temperature (increase by 1 degree) to make it easier for experts to characterize the expected response. As mentioned before, this setup covers inputs relevant for climate change predictions, but it does not represent realistic input changes for climate change. Figure 1 visualizes a simplified expected response of the catchment outlet discharge to these changes discussed in more detail below. The simplified expected responses shown in Fig. 1 represent general trends; the true expected response will be less smooth due to shorter-term precipitation and temperature fluctuations.

1. *Input change*: **Constant relative increase in precipitation by 10%.**

   *Investigated response:* We are interested in the change in discharge at the catchment outlet resulting from the change in precipitation:

$$\Delta Q_P = Q(1.1 \cdot P, T) - Q(P, T) \quad , \tag{1}$$

where $Q$ is the hydrologic model describing catchment outlet discharge as a function of precipitation time series, $P$, and temperature time series, $T$. $\Delta Q_P$ is the change in catchment outlet discharge resulting from the 10% increase in precipitation as predicted by the model. Note that, according to equation (1), such a relative change does not lead to any input change during periods without precipitation. An alternative absolute input modification would not make sense for precipitation as this would lead to the elimination of dry weather periods.

*Expected response:* As shown in the top row of Fig. 1, We expect an increase in catchment outlet discharge that reflects the discharge pattern of the base simulation. Only in cases of short events and considerable travelling of the flood wave, we expect a decrease in discharge at the falling limb of the discharge peak (following an increase at the rising limb),

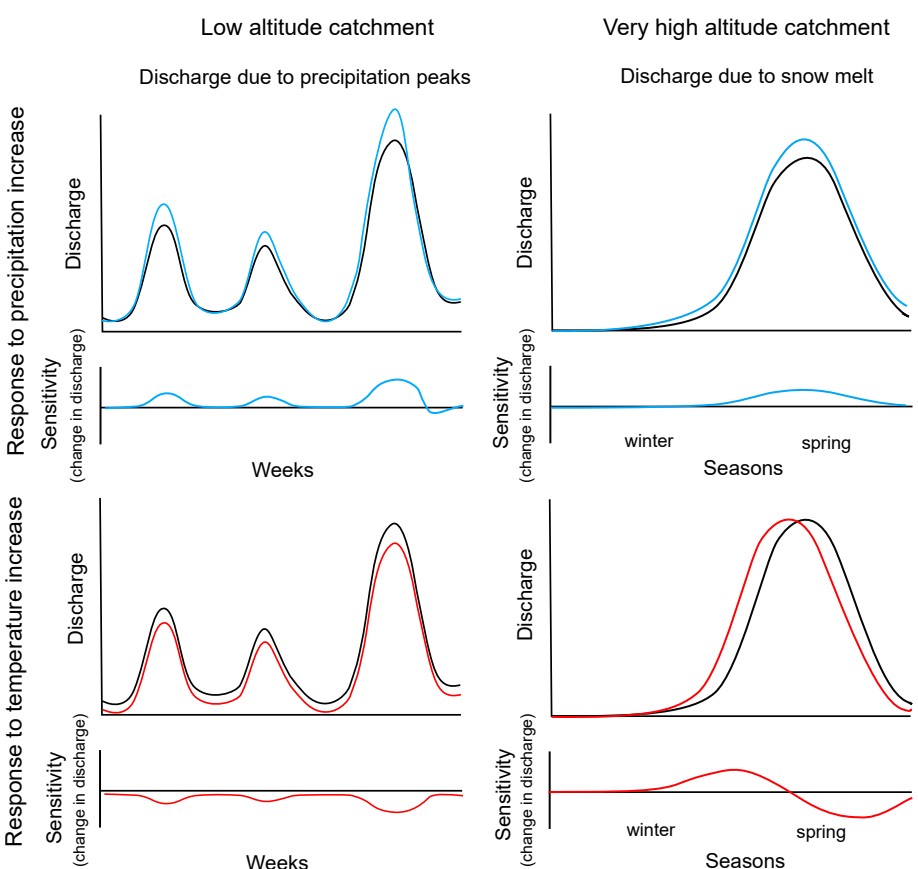

**Figure 1.** Simplified expected response to a precipitation increase by 10% (top row) and to a temperature increase by 1 degree (bottom row) for low altitude catchments (left column; response to precipitation events within weeks) and for high altitude catchments (right column; changes in seasonal snowmelt peak). Black lines: discharge for unmodified input. Blue lines: discharge and sensitivity (change in discharge) for modified precipitation input. Red lines: discharge and sensitivity (change in discharge) for modified temperature input.

due to a shift of the flood peak to earlier times caused by a higher flood wave celerity at higher water levels (Battjes and Labeur, 2017). This expectation is based on the assumption that a 10% increase in precipitation is small enough to not fundamentally change vegetation, soil structure, and other catchment properties. For more complex and stronger input changes, more complex response patterns are possible as discussed by Blöschl et al. (2019). In a world-wide analysis of past trends in water balance and evapotranspiration, Ukkola and Prentice (2013) found some regions (Europe and Canada) with increasing precipitation and decreasing runoff (see Figure 5 in Ukkola and Prentice (2013)). However, as this is an analysis of past data, many other factors changed also, in particular, there was a significant temperature increase in these regions that contributed to increased evapotranspiration whereas we assume no change in temperature for this input change scenario.


2. *Input change:* **Constant increase in temperature by 1°C.**

*Investigated response:* We are interested in the change in discharge at the catchment outlet resulting from the change in temperature:

$$\Delta Q_T = Q(P, T + 1°C) - Q(P, T) \quad , \tag{2}$$

where $\Delta Q_T$ is the change in catchment outlet discharge resulting from the 1°C increase in temperature as predicted by
the model and the other symbols have the same meaning as in equation (1).

*Expected response:* As shown in the bottom left panel of Fig. 1, for warm catchments (without snow cover), we expect a decrease in outlet discharge that is more pronounced in summer than in winter due to increased evapotranspiration. Again, in cases of short events and considerable travelling of the flood wave, we may get a short increase in discharge at the falling limb of the peak (following a decrease at the rising limb) due to a shift of the peak to later times caused
by a lower flood wave celerity at lower water levels (Battjes and Labeur, 2017). For catchments with seasonal discharge pattern dominated by snow cover dynamics, we expect an increase in river discharge in autumn or winter due to a later change of precipitation from rain to snowfall and an earlier melting in spring followed by a decrease in river discharge because the snow melt will be complete earlier. This response pattern is shown in the bottom right panel of Fig. 1. There is less empirical evidence for this expected response in past data (Ukkola and Prentice, 2013) because in most
regions temperature increase is accompanied by precipitation increase and thus leads to increased discharge. However, there are some cases, particularly in North-Asia (see Figure 5 in Ukkola and Prentice (2013)), where there is increase in temperature and runoff despite no significant trend in precipitation. This may be a consequence of a change in snow cover and vegetation.

As the training data contained precipitation or temperature related catchment attributes, such as "mean daily precipitation" and
"fraction of precipitation falling as snow", we compared the results to training with omission of this kind of attributes to avoid biased results due to inconsistent changes in driving forces. Table B1 in appendix B lists the full as well as the reduced sets of catchment attributes used for this comparison.

The intent of our study is to identify potential problems of hydrologic models and to learn from them and not to provide a
representative overview of results of different models. For this reason, we select catchments that allow us to test the response pattern described above as well as possible. As finding reasons for poor fit is a complementary technique of improving models on which we do not focus in this paper, we only select catchments for which all our primary modelling approaches lead to a very good fit during the calibration period (NSE > 0.8 during the calibration period for all investigated model structures; the range of NSE values for the selected catchments was 0.82 - 0.92). All of these models also lead to a good fit during the
validation period (range of NSE values 0.67 - 0.91). To best represent the conditions for which we can describe the expected response as described above, we choose:

– *Low altitude warm basins:*
  These basins should only have a minor amount of snow and thus a relatively simple response patterns as described above.

- *Very high altitude (cold) basins:*

  In these basins, the response should be dominated by the shifts in snowfall and snow melt.

To complement our study, we also chose intermediate altitude catchments:

- *Intermediate altitude basins:*

  For these basins, we expect a combination of the snow-cover-dominated response in winter and spring combined with the warm-basin response in summer. The transition between the two regimes will depend on altitude and latitude, which makes the response less clear than in the other two cases.

## 2.2 Models

### 2.2.1 Conceptual Hydrologic Models

We will compare the conceptual hydrologic model GR4 (Santos et al., 2018), which is a continuous-time version of the model GR4J (Perrin et al., 2003) in combination with a continuous-time version of the snow accumulation model Cemaneige (Valery et al., 2014), which we call "GR4neige", and a continuous-time version of the discrete-time model HBV (Bergström, 1992; Lindström et al., 1997; Seibert, 1999; Seibert and Vis, 2012). All equations of these conceptual hydrologic models are given in Appendix A.

### 2.2.2 Machine Learning Models

The great success of machine learning in hydrology is primarily based on the Long Short-Term Memory (LSTM) models (Kratzert et al., 2018, 2019a, b; Feng et al., 2020). We will thus also exclusively use the LSTM approach to represent machine learning models. The models deviate from each other by their consideration of basins for calibration and by the set of catchment attributes used for calibration (see section 2.3 below). Appendix B provides an overview of the setup of these models.

## 2.3 Calibration/Training

The parameter values of the models were obtained by optimization of a loss function that quantifies the deviation of model output from observations as described below. As it corresponds to the typical use in the literature, for this optimization we use the term "calibration" for the conceptual models and "training" for the machine learning models.

The conceptual hydrologic models used daymet altitude band inputs as provided in the CAMELS data (https://ral.ucar.edu/solutions/products/camels), aggregated to a maximum of 5 bands, for catchment-by-catchment calibration by maximization of the posterior with a simple, uncorrelated, normal error model and wide priors. Optimization was performed using the LBFGS algorithm (Liu and Nocedal, 1989). As there is only incomplete banded input data available for the basins 12167000, 12186000

and 12189500, we calibrated the model only for 668 of the 671 basins of the US CAMELS data set.

The LSTM model was jointly trained to all 671 basins of the US CAMELS data set (https://ral.ucar.edu/solutions/products/camels) using daymet forcing and the catchment attributes listed in Table B1 in the Appendix (Newman et al., 2015; Addor et al., 2017) and maximizing the Nash-Sutcliffe Efficiency (NSE). Optimization was performed using the AdaDelta Optimizer with parameters lr = 1.0 and rho = 0.9 (Zeiler, 2012). As we encountered some unexpected responses in the low-altitude basins to a change in temperature (see section 3.2.1 below), additional trainings were done, as described in section 3.3.


In both cases, we used the same 15 years for calibration and the same 15 years for validation as in the original publication by Newman et al. (2015) (1980/10/01-1995/09/30 for calibration and 1995/10/01-2010/09/30 for validation).

## 2.4   Implementation

The conceptual hydrologic models were implemented in Julia (Bezanson et al., 2012, 2017) using the packages DifferentialEquations.jl (Rackauckas and Nie, 2017), ForwardDiff (Revels et al., 2016), and Optim (Mogensen and Riseth, 2018).

The LSTM was implemented in Python (Van Rossum and Drake, 2009) using Pytorch (Paszke et al., 2019).

All our code is publicly available (Conceptual models: https://doi.org/10.25678/000CQ0. LSTM: http://doi.org/10.5281/zenodo.3993880).

## 3   Results and Discussion

### 3.1   Quality of Fit

Fig. 2 provides an overview of the Nash-Sutcliffe Efficiency (NSE) values achieved for the calibration and validation periods
for all modelling approaches, for the 668 basins for which also the conceptual models could be calibrated as well as for the 12 basins selected for metamorphic testing (see next section). These results clearly confirm the strength of the LSTM model compared to the conceptual hydrologic models regarding the quality of fit for calibration as well as validation periods. The LSTM has the additional advantage that it generalizes very well to catchments not used for training but this feature is not investigated in this paper.

### 3.2   Metamorphic Testing

For metamorphic testing, we separately evaluated basins that belong to the three classes of low altitude, warm basins, very high altitude basins, and intermediate altitude basins mentioned in section 2. For each of the three classes, we selected four basins.

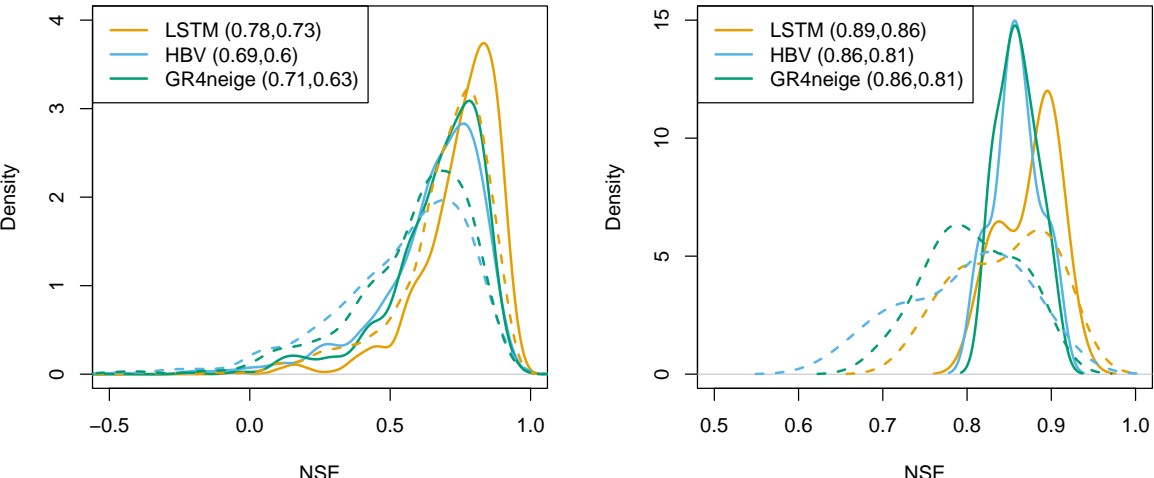

**Figure 2.** Overview of NSE values of all modelling approaches for the calibration period (solid) and the validation period (dashed) for all 668 basins (left) and for the 12 basins selected for metamorphic testing (right; note the different scale of the x-axis). The median NSE values are indicated in brackets in the legend (calibration period, validation period).

Fig. 3 provides an overview of the locations of the selected four basins within each category. As mentioned in section 2.1 these basins were selected by allowing for an excellent fit for all modelling approaches (NSE > 0.8 during the calibration period; the range of NSE values across models and selected catchments was 0.82 - 0.92 for the calibration period and 0.67 - 0.91 for the validation period). Due to the limited number of basins in these categories, the strong requirement regarding the quality of fit for all modelling approaches, and the wish to have the same number of basins in each category, it was not possible to compare more basins. However, as shown in the following sections, there are quite consistent patterns of responses to changes within each of these categories.

### 3.2.1 Low Altitude Warm Basins

Figure 4 shows the results for the final year of the calibration period for a typical warm, low altitude basin. Results for more years during the calibration and validation periods and for more low altitude basins are provided in the Figures SI.2 to SI.17 in the Supporting Information. These results are systematic across all studied basins, demonstrating that the features discussed in this section represent the typical behavior of this kind of basins and are not just an artifact of this specific basin and year.

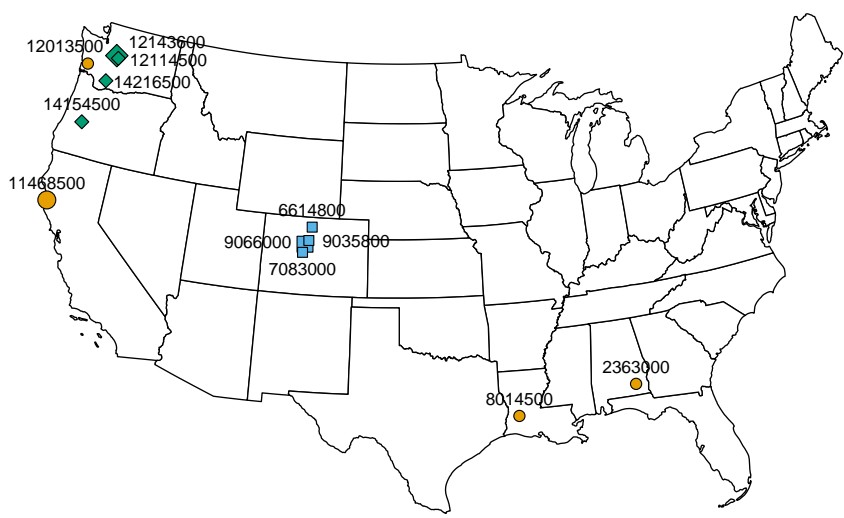

**Figure 3.** Basins used for metamorphic testing. Orange circles indicate low altitude basins, blue squares very high altitude basins, and green rhombs intermediate altitude basins. The large markers represent basins with results shown in the main paper, the results for all basins are shown in the Supporting Information. The numbers represent CAMELS basin identifiers. (Map produced with the R-package `usmap`: https://CRAN.R-project.org/package=usmap).

As is shown by the NSE values in the legends of the fourth panels (Fig. 4 and Figs. SI.2 to SI.17), for these basins, all of the compared primary modelling approaches (GR4neige, HBV, LSTM) provide an excellent fit over the calibration and validation periods (all NSE values are larger than 0.8 during calibration and larger, mostly much larger, than 0.65 during validation, see also the overview of NSE values in Fig. 2).

The sensitivities to a 10 % increase in precipitation, $\Delta Q_P$ (see equation 1), are plotted in the top panel of Figure 4 (and of Figs. SI.2 to SI.17). All our modelling approaches (GR4neige, HBV, LSTM) lead to very similar sensitivities to the investigated relative change in precipitation. The sensitivities to the investigated increase in precipitation also correspond to our expectations as described in section 2.1 (see in particular Fig. 1, top left panel), as they are positive and larger during precipitation events than during dry weather periods (compare time series of the precipitation sensitivities in the top panel to the time series of discharge in the bottom panel). The result of this metamorphic test therefore belongs to the category (A) outlined in section 2.1 (consistent agreement with expectations across modelling approaches) and makes us confident into the response of all models to changes in precipitation.

In contrast to the precipitation sensitivities shown in the first panel, the second panel of Figure 4 (and of Figs. SI.2 to SI.17) shows substantial differences in temperature sensitivities, $\Delta Q_T$ (see equation 2), between different modelling approaches

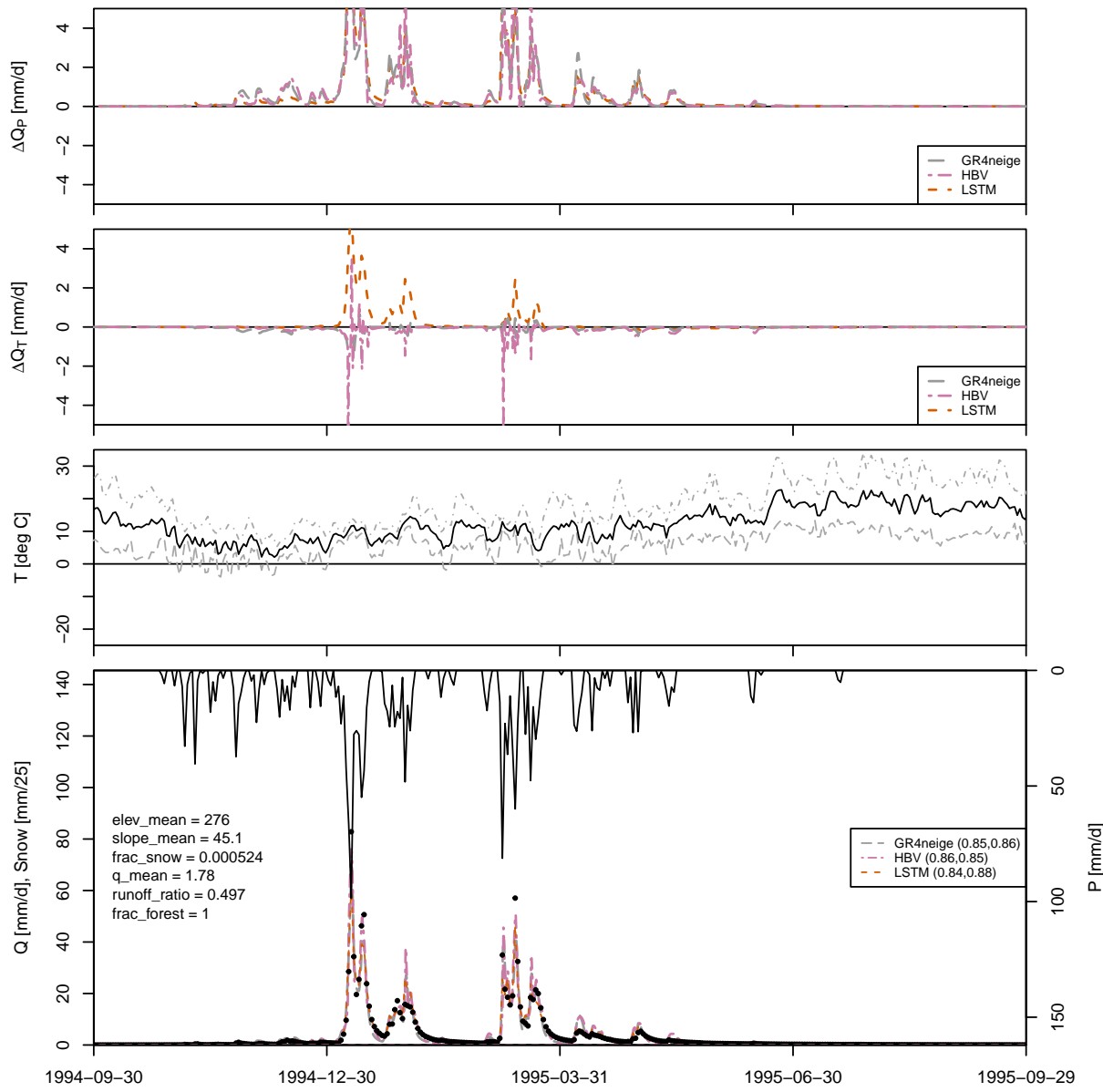

**Figure 4.** Results for basin 11468500. First panel: modelled sensitivity of discharge to a 10% increase in precipitation, $\Delta Q_P$ (see equation 1). Second panel: modelled sensitivity of discharge to a 1 degree increase in temperature, $\Delta Q_T$ (see equation 2). Third panel: minimum, mean and maximum temperature. Fourth panel: observed precipitation (from top, right axis); modelled (lines) and observed (circles) discharge and modelled snow cover in max. five altitude bands (dashed lines, left axis, zero for this particular basin), NSE for calibration and validation periods in brackets in the legend; and the values of selected catchment attributes according to Addor et al. (2017) (on the left).

(GR4neige, HBV, LSTM). The sensitivities of the hydrologic models GR4neige and HBV are essentially negative (the discharge for increased temperature is smaller than it was with the original temperature) with only some brief positive excursions associated with small shifts in discharge peaks. These are the expected sensitivities as described in section 2.1 (see in particular Fig. 1, bottom left panel). In contrast, the LSTM often shows a positive response of catchment outlet discharge to the investigated temperature increase, in particular during flood events. This seems to be an implausible response, as increased temperature increases evaporation whereas precipitation does not change in our metamorphic testing scenario. The result of this metamorphic test thus belongs to category (C), outlined in section 2.1 (inconsistency with expectations for some model structures). This raises the question of which approach may provide the correct response. The conceptual hydrolgical models may share similar deficits as the expected response as both are based on similar expert knowledge. On the other hand, the LSTM may, due to its broad coverage of climatic conditions of 671 Camels basins, better consider the effect of changing catchment properties resulting from the increasing temperature or its response may be incorrect due to poor out-of-sample prediction. Since we see here a striking difference in the behaviour of models that fit and predict very well under current climatic conditions, we have to investigate how consistent the response of the LSTM is across different training options. This can provide additional hints regarding which of the two explanations discussed above may be more plausible. This will be investigated in section 3.3.

### 3.2.2 Very High Altitude (Cold) Basins

Figure 5 shows the results for the final year of the calibration period for a typical very high altitude, cold basin. Results for more years during the calibration and validation periods and for more high altitude basins are provided in the Figures SI.18 to SI.33 in the Supporting Information. These results demonstrate that the features discussed in this section represent the typical behaviour of this kind of basins and are not just an artifact of this specific basin and year.

The legends of panels 4 in these figures (Fig. 5 and Figs. SI.18 to SI.33) show again that we have an excellent fit during the calibration as well as validation periods for all modelling approaches (GR4neige, HBV, LSTM) with NSE values larger than 0.8 during calibration and larger than 0.7 during the validation periods.

The precipitation sensitivities show in this case more differences than for the low altitude catchments in section 3.2.1. All models show the expected positive precipitation sensitivities (higher discharge for higher precipitation), but the response of the LSTM is considerably smaller and smoother than the responses of the conceptual models. Still, these results correspond qualitatively to our expectations as described in section 2.1 (see in particular Fig. 1, top right panel).

Also the temperature sensitivities show the expected behavior of a positive sensitivity (higher discharge for higher temperature) due to the earlier snow melt process followed by a negative sensitivity (lower discharge for higher temperature) due to the earlier completion of the snow melt process (see section 2.1, in particular Fig. 1, bottom right panel)). There is a tendency that the positive response starts later and the negative response ends earlier for the LSTM compared to the conceptual hydro-

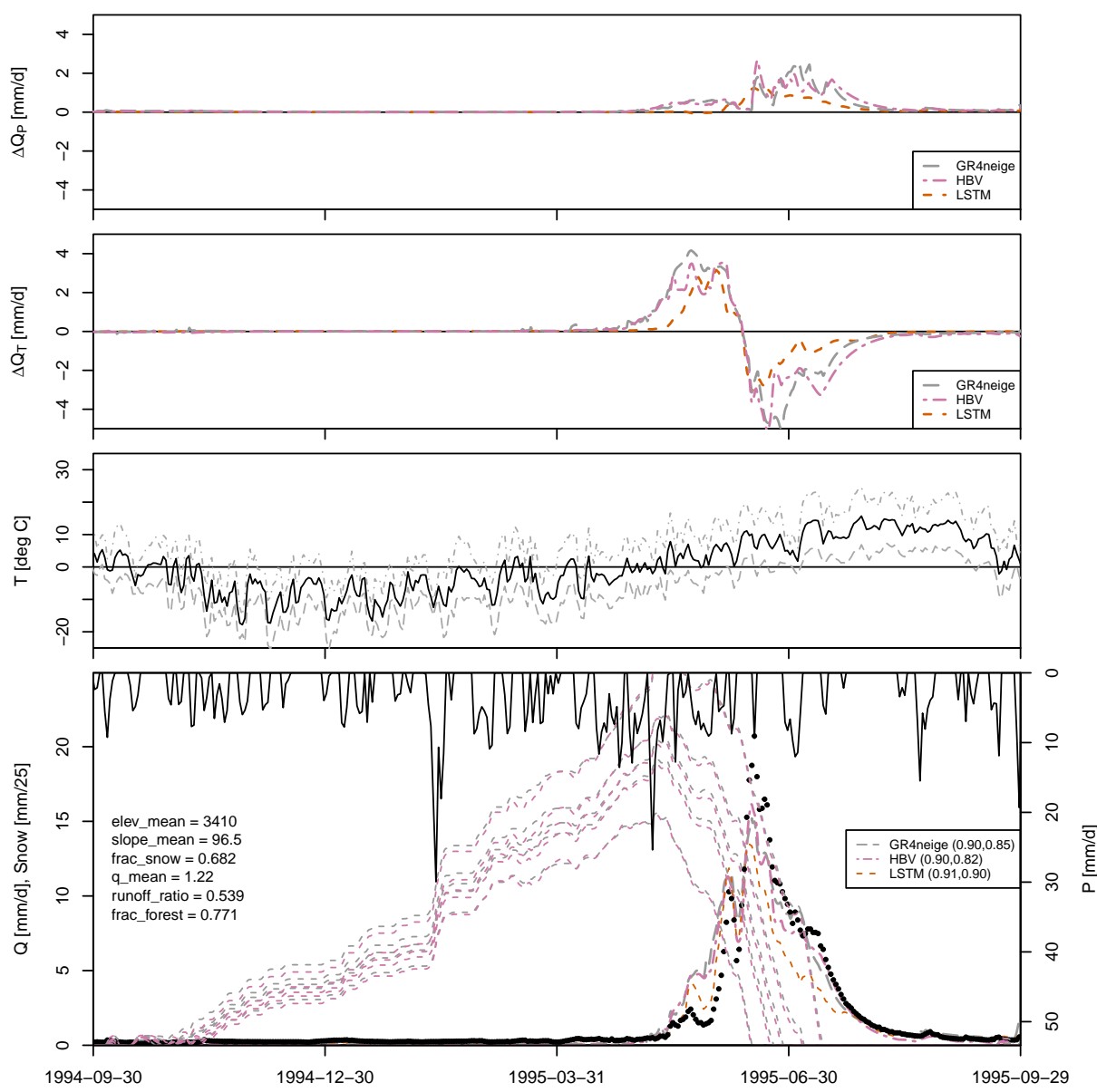

**Figure 5.** Results for basin 09066000. First panel: modelled sensitivity of discharge to a 10% increase in precipitation, $\Delta Q_P$ (see equation 1). Second panel: modelled sensitivity of discharge to a 1 degree increase in temperature, $\Delta Q_T$ (see equation 2). Third panel: minimum, mean and maximum temperature. Fourth panel: observed precipitation (from top, right axis); modelled (lines) and observed (circles) discharge and modelled snow cover in max. five altitude bands (dashed lines, left axis), NSE for calibration and validation periods in brackets in the legend; and the values of selected catchment attributes according to Addor et al. (2017) (on the left).

logic models. Also these sensitivities tend to be smaller and smoother for the LSTM than for the conceptual hydrologic models.

The results of these metamorphic tests thus belong to the category (B) outlined in section 2.1 (significant differences between approaches, but still in qualitative agreement with the expectations). As mentioned in section 2.1, this result shows the limits of metamorphic testing, as it is difficult to judge which of the quantitative responses is closer to reality. Nevertheless, metamorphic testing with multiple models demonstrates that models that provide a similarly good fit during calibration and validation periods can still differ considerably in their response to modified driving forces. This indicates to be cautious with predictions of such responses.

### 3.2.3 Intermediate Altitude Basins

Figure 6 shows the results for the final year of the calibration period for a typical intermediate altitude basin. Results for more years during the calibration and validation periods and for more intermediate altitude basins are provided in the Figures SI.34 to SI.49 in the Supporting Information. These results demonstrate that the features discussed in this section represent the typical behavior of this kind of basins and are not just an artifact of this specific basin and year.

The legends of panels 4 in these figures (Fig. 6 and Figs. SI.34 to SI.49) show again that we have an excellent fit during the calibration as well as validation periods for all modelling approaches (GR4neige, HBV, LSTM) with NSE values larger than 0.8 during calibration and larger than 0.77 during the validation periods.

The results shown in Fig. 6 combine the results discussed in the previous sections, but resemble more closely the high altitude catchments, as still snow cover dominates the dynamic behavior during most of the season.

The panels 1 and 2 of Fig. 6 show clearly that over the first half of the considered period, all models show very similar responses with respect to the change in precipitation as well as to the change in temperature. In contrast, in the second half of the year, the conceptual models agree with one another but deviate from the LSTM. In this part of the season, the responses of the LSTM is smoother and smaller than that of the hydrologic models. Again, the qualitative nature of metamorphic testing makes it difficult to assess which of these results are more plausible. These results are again in category (B) of our results classification for metamorphic testing outlined in section 2.1.

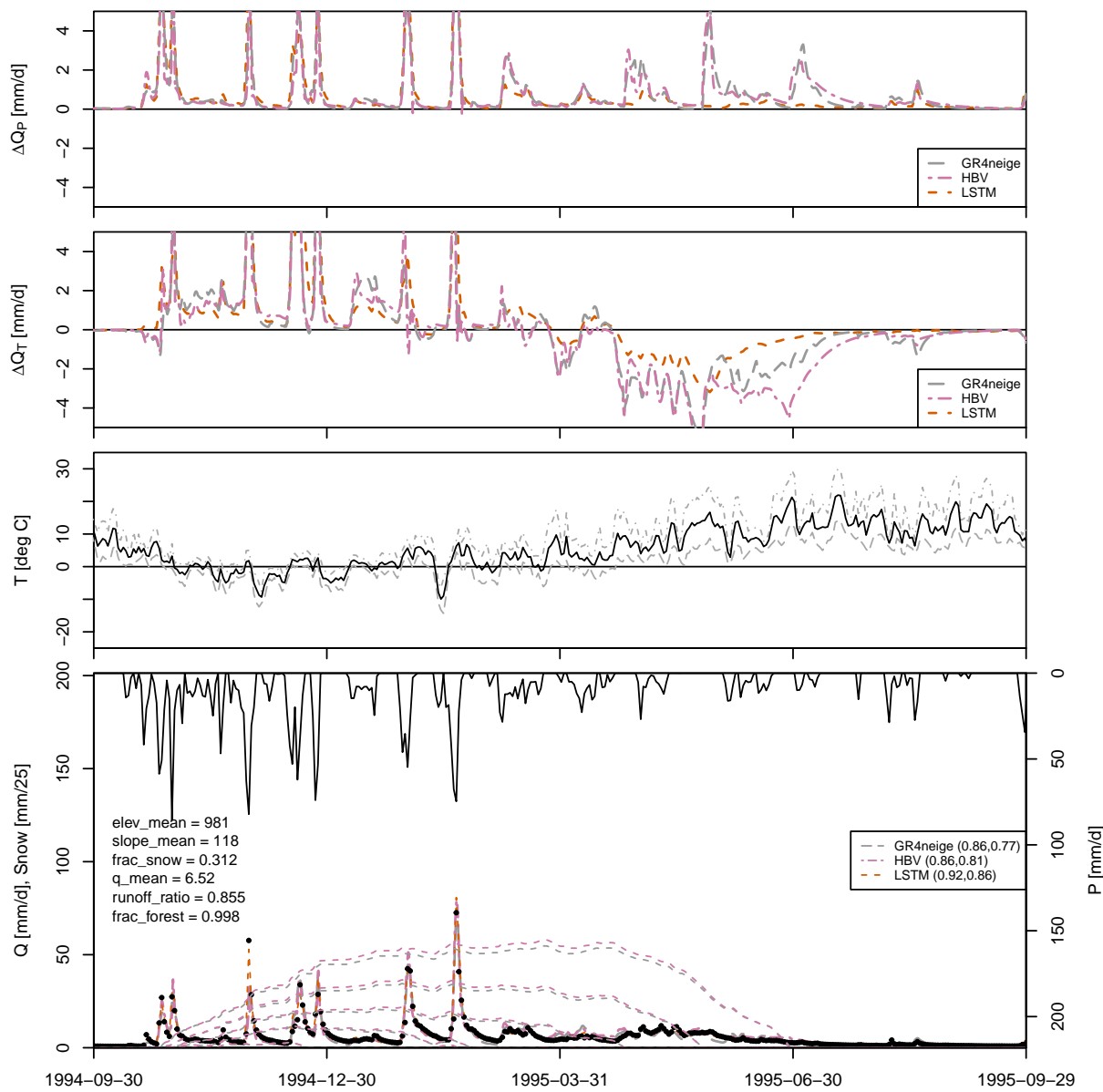

**Figure 6.** Results for basin 12143600. First panel: modelled sensitivity of discharge to a 10% increase in precipitation, $\Delta Q_P$ (see equation 1). Second panel: modelled sensitivity of discharge to a 1 degree increase in temperature, $\Delta Q_T$ (see equation 2). Third panel: minimum, mean and maximum temperature. Fourth panel: observed precipitation (from top, right axis); modelled (lines) and observed (circles) discharge and modelled snow cover in max. five altitude bands (dashed lines, left axis), NSE for calibration and validation periods in brackets in the legend; and the values of selected catchment attributes according to Addor et al. (2017) (on the left).

### 3.3 Sensitivity to Attributes, Calibration Set, and Seed for Low Altitude, Warm Basins

As the results for the temperature sensitivities for the low altitude, warm basins were most striking, showing most of the time different signs for the LSTM than for the conceptual hydrologic models, we tried to learn more about the reasons for this phenomenon. To investigate this problem, we performed a sensitivity analysis of the LSTM model regarding

   – catchment attributes considered for training,

   – basins considered for training, and

– seed of the random number generator that affects the local minimum found by the optimizer.

The idea of using fewer catchment attributes was an attempt to improve the representation of physical processes by the LSTM by only allowing the use of the attributes with a direct physical influence (e.g. omitting mean elevation as temperature has a dominant influence on the physical processes whereas elevation is much less relevant, but could be used as a proxy for temperature by the LSTM) and removing attributes that would have to be modified for prediction with modified input (e.g. all

precipitation-related attributes, such as mean daily precipitation, as this information should be inferred from the precipitation time series and would have to be adjusted when modifying precipitation input). The motivation for reducing the set of training basins was to reduce the diversity of basins and primarily keep basins with low elevation (and still sufficient diversity within this class). Finally, the test with different seeds was motivated by checking whether the results where caused by converging into a "bad" minimum, whereas other local minima would lead to better results. Table 1 lists the model versions used for this

| Model | Description |
|---|---|
| GR4neige | conceptual model GR4neige as described in appendix A2 |
| HBV | conceptual model HBV as described in appendix A3 |
| LSTM | LSTM as described in appendix B trained with all 671 basins |
| LSTM_red | LSTM trained with all 671 basins using only catchment attributes marked with "x" in Table B1 |
| LSTM_515 | LSTM trained with the 515 basins with a mean altitude < 1000 m (five different seeds) |
| LSTM_361 | LSTM trained with the 361 basins with a mean altitude < 500 m (five different seeds) |
| LSTM_211 | LSTM trained with the 211 basins with a mean altitude < 300 m (five different seeds) |

**Table 1.** Overview of models. The first three rows describe the basic models planned to be used in the project, the lower four rows are the additional model versions used for the sensitivity analysis to analyze the problem of the deviating temperature sensitivities of the LSTM for low altitude catchments.

sensitivity analysis.

Figure 7 shows the precipitation and temperature sensitivities of these models at a higher scale and for a shorter time period than in Figure 4 to facilitate the distinction of the larger number of curves. Results for more years during the calibration and

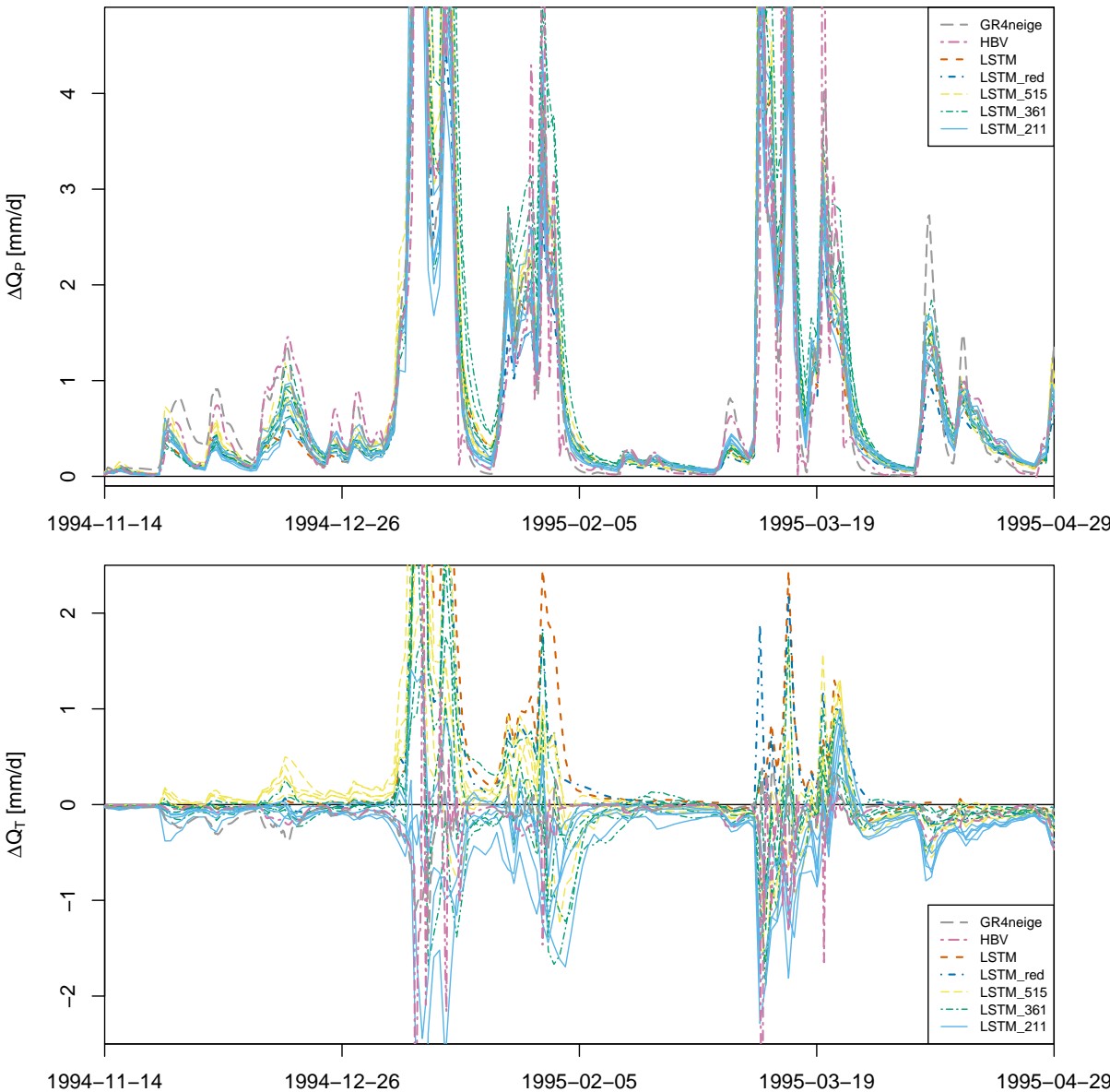

**Figure 7.** Sensitivities for basin 11468500 and different LSTM calibration options (see Table 1). The sensitivities for the models GR4neige, HBV and LSTM are the same as those shown in Figure 4. In addition, LSTM_red shows the sensitivities when calibrating with a reduced set of catchment attributes, and LSTM_515, LSTM_361 and LSTM_211 show the sensitivities when calibrating the LSTM model with different sets of low-elevation basins. For these three cases, results for 5 different seed values are shown. See text for more details.

validation periods and for more low altitude basins are provided in the Figures SI.50 to SI.65 in the Supporting Information.

The results are qualitatively similar throughout all catchments and periods.

     The precipitation sensitivities, $\Delta Q_P$ (see equation 1), are quite insensitive to any of these modifications from the original setup (see upper panel in Fig. 7 and in Figs. SI.50 to SI.65). In particular, it is remarkable that omitting the mean precipitation (that had not been changed when increasing the input precipitation time series) from the input does not change the results.

This indicates that the response pattern for precipitation change is determined from the input time series rather than from this specific catchment attribute. Also reducing the training data set and changing the random seed does not change the observed precipitation sensitivities.

     The results for the temperature sensitivities, $\Delta Q_T$ (see equation 2), are quite different for the different modelling approaches

(see lower panel in Fig. 7 and in the Figs. SI.50 to SI.65). In particular, those of LSTM and LSTM_515 are mostly positive, whereas those of LSTM_211 (trained only with the basins with mean elevation smaller than 300 m) are negative except for a single peak after the tick mark for March 19, 1995. The sensitivities of LSTM_361 are less visible due to the congestion in the figure, but the trend from LSTM to LSTM_211 is very clear.

To better visualize the differences between the temperature sensitivities of all modelling approaches, Fig. 8 shows a quantification of these differences. As in metamorphic testing the quantitative change in results is difficult to assess, we focus on the quantification of differences in sensitivities for which the signs of the sensitivities are different between different model versions. We therefore calculated the mean squared differences in sensitivities of all combinations of approaches, setting the differences to zero if the sensitivities have the same signs and omitting periods of very low sensitivities ($< 0.1$ mm/d). Of

these values we took the square root and the mean across all basins of the same class (here low altitude catchments) and across different random number seeds. These results are shown in Fig. 8. The results in the first two columns clearly show that the sensitivities of the LSTM model are considerably different from those of the GR4neige and HBV models and that they approach the results of these models when moving from LSTM over LSTM_red, LSTM_515, LSTM_361 to LSTM_211 (see Table 1 for model definitions). In parallel, in this order, the sensitivities deviate more and more from those of the original LSTM model

(third column in Fig. 8). These quantitative difference measures thus clearly confirm the qualitative discussion in the previous paragraph.

     The same quantification as shown for the temperature sensitivities in Fig. 8 was performed for precipitation and temperature sensitivities and is shown in Fig. SI.1 in the Supporting Information. These results clearly demonstrate that there is no similar

problem in the signs of calculated sensitivities for precipitation for all classes of basins and for temperature for the very high elevation basins. The temperature sensitivities for the intermediate elevation basins also show problems between the modelling approaches, but these are more difficult to interpret as they combine effects for low and high altitude catchments.

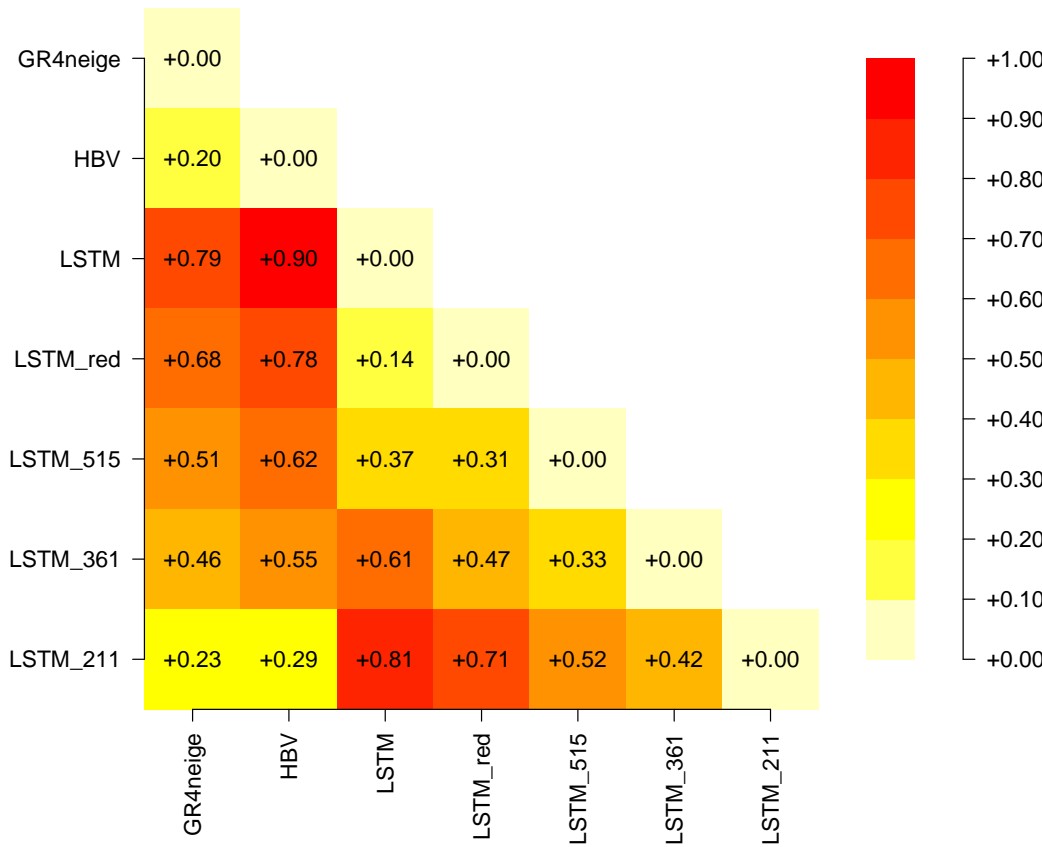

**Figure 8.** Differences between the temperature sensitivities $\Delta Q_T$ of all modelling approaches quantified as described in the text (mm/d).

Table 2 summarizes the results of the extended metamorphic testing. Our sensitivity analysis demonstrates that for the
LSTM applied to the low altitude catchments there is a large sensitivity to the training set of modelled catchment outlet
response resulting from input temperature changes. It is particularly remarkable that the sensitivities to temperature change
are different despite all of the different calibration versions provide an excellent fit during calibration and validation periods.
This suggests that the problem of making predictions for new environmental conditions, which is very relevant *e.g.* for climate
predictions, cannot be detected from only comparing the quality of fit during calibration, validation, and for catchments not
used for calibration, as it is has been done in most studies so far. As the responses of LSTM_211 are mostly in qualitative
agreement with the expected responses, it seems plausible that the deviations shown for the other calibration options from the
expected response are not caused by an error in the expert opinions, but are rather a question of LSTM calibration. This result
seems to indicate that the LSTM trained with all basins is not as universal as expected from previous results, but that an LSTM

| Model | low altitude | | high altitude | | intermed. alt. | |
|---|---|---|---|---|---|---|
| | prec. | temp. | prec. | temp. | prec. | temp. |
| GR4neige | A | A | B | B | B | B |
| HBV | A | A | B | B | B | B |
| LSTM | A | **C** | B | B | B | B |
| LSTM_red | A | **C** | B | B | B | B |
| LSTM_515 | A | **C** | | | | |
| LSTM_361 | A | **C** | | | | |
| LSTM_211 | A | A | | | | |

**Table 2.** Summary of results of metamorphic testing. See section 2.1 for a description of the result categories (note that the lowest three models were designed for low-altitude catchments and were therefore not used for intermediate and high altitude catchments).

trained more specifically with the low altitude basins passes the metamorphic test. It is possible that pre-conditioning the model to these catchments could have a similar effect.

## 3.4 Discussion

Our results uncovered problems of LSTM models to produce consistent (primarily across training data) results for predictions to input changes beyond those used during training. These results provide a motivation for investigating approaches that either (A) widen the diversity of basins used for training to expand the potential for learning physical mechanisms (Wi and Steinschneider, 2022), or (B) to try to combine the strengths of machine learning and conceptual (or even physical) hydrologic models (Ng et al., 2023; Shen et al., 2023; Tsai et al., 2021; Jiang et al., 2020). As approach (A) failed for the low level basins in our study (when extending the training data set to high elevation basins), our results indicate that approach (B) seems more promising for systems that we have a good mechanistic knowledge of. Examples of such hybrid approaches are considering physical constraints or mechanisms in machine learning models (Nearing et al., 2020; Razavi, 2021; Xie et al., 2021; Zhong et al., 2023), postprocessing the output of mechanistic models with machine learning models (Konapala et al., 2020), using conceptual model output or components (such as evapotranspiration estimates) as additional input to machine learning models (Wi and Steinschneider, 2022), or inferring functional relationships in conceptual hydrologic models by replacing parameterized elements or functions by machine learning models (Jiang et al., 2020; Tsai et al., 2021; Höge et al., 2022). In the last category, which some referred to as differentiable modeling (Shen et al., 2023), neural networks are seamlessly connected to programmatically differentiable (permitting gradient tracking) process-based equations and they are trained together in an end-to-end fashion (Jiang et al., 2020; Feng et al., 2022; Bindas et al., 2024). This framework may address the sensitivity problem by hardcoding (and thus guaranteeing) required physical sensitivities to forcings and attributes as prior equations, or restricting information inflow into and out of neural networks, which should be investigated in the future. If sensitivity is a primary concern, one should also use caution with neural networks as postprocessing layers as they can modify the assumed

sensitivities.

These results lead thus to the following conclusions regarding the questions raised in the introduction:

**A good fit during calibration and validation periods does not guarantee a good response to changes in driving forces.** There is a strong need for analyzing model predictions beyond the quality of fit (usually quantified by NSE) and to compare
predictions for different model structures to gain confidence in predictions and to gain insight into model prediction uncertainty. This need is evident as we demonstrated that models that fit similarly well for calibration and validation periods can still show strong differences in their response to modified inputs. Good fits during calibration and validation periods are thus not a sufficient criterion for a model to predict the response to modified driving forces accurately. This is a very important conclusion to keep in mind when using hydrologic models of climate change prediction.

**Usefulness of metamorphic testing.** Metamorphic testing is a very useful tool to test models beyond the usual calibration-validation process. The problem of metamorphic testing is that it requires the response to be at least qualitatively known. This can be difficult and even biased as this requires input of expert knowledge that can be biased by the current state of incomplete scientific knowledge. For this reason, we strongly recommend to go for "extended metamorphic testing" in which we do not only check model predictions for modified inputs with the expected response but in a multi-model
approach we also investigate the sensitivity of these results to the model structure and to training data, and algorithmic parameters. This extended analysis can uncover "objective problems" such as a dependence of the response (in our case study even the sign of the response) on choices of the training data, which clearly indicates a problem that is not dependent on the partly subjective prescription of the expected response. It can also, and did in our case study, uncover quantitative differences in responses between different model structures even in cases in which all models passed the
qualitative metamorphic test (see the large numbers of "B" classifications listed in Table 2).

**Using more data for training can be deleterious.** Our results seem to be in contradiction to the general principle that machine learning models always profit from the extension of the training data set. In our case study, adding high elevation basins to the training data set does not reduce the quality of fit but it deteriorates the response of low altitude basins to temperature change. This demonstrates that adding data that is not directly relevant for a specific prediction (in our case
for low elevation basins) can have an adverse effect. On the other hand, when we further reduced the training data set, the quality of fit and prediction deteriorated (not shown in the paper). For this reason, this is not a contradiction to the statement that adding "useful data" - data that provides information directly relevant for the question to be investigated - improves the quality of fit and the response to input changes. However, it may raise the awareness for carefully selecting training data as adding less relevant data (for this specific question) may have adverse effects.

**Machine Learning vs Conceptual Models.** The modelling approaches based on machine learning and on conceptual hydrologic models have complementary strengths and deficits. Machine learning models are particularly strong in providing an excellent quality of fit and prediction accuracy for validation periods as well as for the prediction of ungauged catchments. However, they need to be calibrated to a large set of basins and their performance can be poor when calibrated

to a single basin. In addition, we provide examples in which the responses of machine learning models to changes in driving forces are very sensitive to the basins selected for training and can be implausible. On the other hand, conceptual hydrologic models need much less data, in particular they can easily be applied to a single basin. They generally provide an inferior fit during calibration and validation periods but seem to show a more plausible and more consistent response to changes in driving forces beyond those present in the calibration data set. However, whenever input changes are strong enough to alter catchment properties, such as vegetation or soil structure, prediction with conceptual models becomes unreliable unless the required modifications to their parameters would be known or vegetation and soil structure would be part of the model. This would lead to a model with mechanisms that are very difficult to parameterize with sufficient accuracy. In principle, machine learning models could be better for such predictions, as they could learn the effect of such changes in catchment properties from other catchments. However, as we have seen already for relatively small changes in driving forces, more research is needed to realize this potential. This would need the development of models that are trained on a training data set that contains a large diversity of catchments with different characteristics and/or that are constrained or preconditioned by physical and biological considerations.

## 4 Summary and Conclusions

We compared the sensitivity of runoff predictions of conceptual and machine learning hydrologic models to changes in precipitation and temperature input for selected catchments of the CAMELS-US data set for which all modelling approaches provided a good fit for the calibration and validation time periods. We found the following results:

- We confirmed earlier results by various researchers that machine learning models provide generally a better fit (higher NSE) for both calibration and validation periods. In addition, machine learning models are much better in extrapolating to basins not used for calibration, but this was not the main aim of this study.

- In an extended metamorphic testing setup, we found qualitatively similar responses of the catchment outlet discharge to precipitation and temperature changes for intermediate- and high-elevation basins, with the main quantitative difference that the responses of the LSTM were generally smaller and smoother than those of the conceptual hydrologic models. As metamorphic testing is a qualitative procedure, it is hard to assess which of these responses are more plausible. On the other hand, we found major differences in the responses for low altitude basins for which the LSTM models led to less plausible results (positive rather than negative responses of catchment outlet discharge to a temperature increase). Training the LSTM with a reduced set of catchment attributes, that should represent the direct physical influence factors, did not resolve this issue. However, training the LSTM on only low-elevation catchments reversed the sign of the sensitivities that then mostly agreed with that of the conceptual hydrologic models. As for the intermediate- and high-elevation basins, the response of the LSTM model was then in qualitative agreement with that of the conceptual models but generally smaller and smoother.

Our results indicate the need for caution with the prediction of LSTM models for inputs that were not present in a similar form in the training data set. Enlarging the training set to situations that are not of direct relevance for the investigated problem may even deteriorate the results. In our case study, this occurred with results for low altitude basins when also including high altitude basins for training. On the other hand, enlarging the set of low altitude basins improved the response for low altitude basins, in agreement with the experience with machine learning models that a large training set of basins is important for

leveraging their full potential. These results provide a motivation for intensifying research regarding approaches that try to combine the strengths of machine learning and conceptual (or even physical) hydrologic models. Hybrid approaches that profit from physical constraints and machine learning flexibility could eliminate the problem of implausible behavior and reduce the sensitivity to the training data set of the LSTMs and, on the other hand, improve the quality of fit compared to the conceptual hydrologic models.

## Appendix A: Conceptual Hydrologic Models

### A1 Auxiliary Functions

We introduce here auxiliary functions that are used to smooth transitions between different hydrologic regimes. Smooth transitions lead to smoother posterior shapes, facilitate numerics, and are more realistic even in cases of physically sharp transitions as they represent averages over the catchment where the environmental conditions that determine the transitions are not homogeneous Kavetski et al. (2006).

We suggest two parameterizations of a smoothed Heaviside function:

$$f_{\mathrm{SH}}^{\mathrm{logistic}}(x, \Delta x) = \frac{1}{1 + \exp\left(-4\dfrac{x}{\Delta x}\right)} \tag{A1a}$$

$$f_{\mathrm{SH}}^{\mathrm{quadratic}}(x, \Delta x) = \begin{cases} 0 & \text{for } x \leq -\Delta x \\ \dfrac{1}{2}\left(\dfrac{x + \Delta x}{\Delta x}\right)^2 & \text{for } -\Delta x < x \leq 0 \\ 1 - \dfrac{1}{2}\left(\dfrac{\Delta x - x}{\Delta x}\right)^2 & \text{for } 0 < x \leq \Delta x \\ 1 & \text{for } x > \Delta x \end{cases} \tag{A1b}$$

These functions are visualized in Figure A1. The two shapes are very similar, but note that the quadratic version is exactly

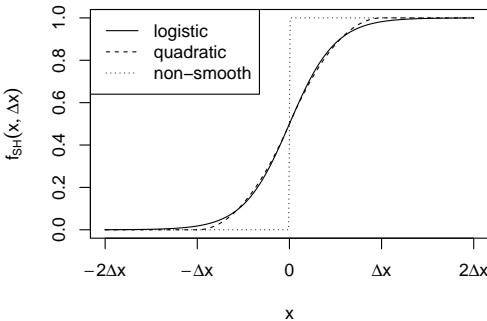

**Figure A1.** Shapes of the smoothed Heaviside functions.

zero or unity for $x \leq -\Delta x$ or $x \geq \Delta x$, respectively, whereas the logistic version approaches these values asymptotically.

The smooth transition function from zero to a linear increase is given by the equation

$$f_{\text{SI}}^{\text{quadratic}}(x, \Delta x) = \begin{cases} 0 & \text{for } x \leq -\Delta x \\ \dfrac{(x + \Delta x)^2}{4\Delta x} & \text{for } -\Delta x < x \leq \Delta x \\ x & \text{for } x > \Delta x \end{cases} \tag{A2}$$

and is visualized in Figure A2. Note that the function exactly matches its non-smooth version for $x \leq -\Delta x$ and for $x \geq \Delta x$.

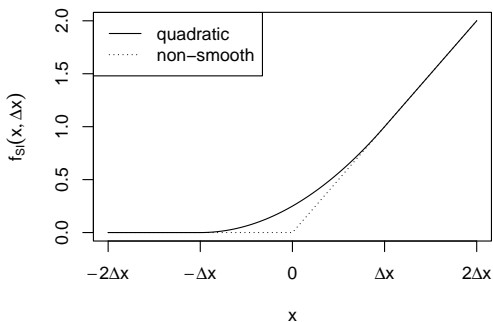

**Figure A2.** Shape of the smoothed start of linear increase function.


These two functions will be used in addition to exponential functions to formulate smooth transitions in the conceptual hydrologic models.

## A2 GR4neige

The GR4J model is a conceptual hydrologic model formulated with a daily time step (J = journellement = daily) that has proven to lead to an excellent performance when only four parameters (thus the 4 in the name) are fitted for a given catchment Perrin et al. (2003). As our objective is to simulate in continuous time, we use the continuous-time version GR4 Santos et al. (2018).

As the set of catchments extends to high altitudes, we extend the continuous-time version of the GR4J model Santos et al.
(2018) by a continuous-time version of the discrete time snow accumulation model "Cemaneige" Valery et al. (2014). We thus call this model "GR4neige" to refer to the original models. Our notation is a compromise between the original publications and the attempt to use similar parameter names across different models. Figure A3 gives a schematic overview of the model.

To formulate the snow model, the catchment is divided into $n_b$ elevation bands for which precipitation and temperature input is required.


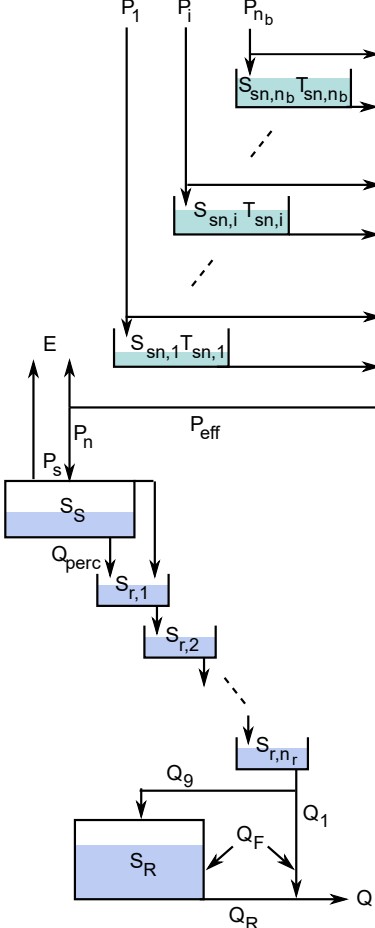

**Figure A3.** Schematic diagram of the GR4neige model (Santos et al., 2018; Valery et al., 2014, modified).

Precipitation is divided into snow and rain by using the fraction of precipitation calculated from daily minimum and maximum temperature as follows:

$$
f_{\text{snow},i} = \begin{cases} 0 & \text{for } T_{\text{min},i} \geq T_{\text{sf,th}} \\ \dfrac{T_{\text{sf,th}} - T_{\text{min},i}}{T_{\text{max},i} - T_{\text{min},i}} & \text{for } T_{\text{max},i} > T_{\text{sf,th}} \text{ and } T_{\text{min},i} < T_{\text{sf,th}} \\ 1 & \text{for } T_{\text{max},i} \leq T_{\text{sf,th}} \end{cases} .
\tag{A3}
$$

Here, the index $i$ refers to the elevation band, $T_{\text{min},i}$ and $T_{\text{max},i}$ refer to the daily minimum and maximum temperature in the elevation band $i$, and $T_{\text{sf,th}}$ is the threshold temperature for snowfall (see Table A1 for a list of all model parameters and their default values and ranges).

The snowpack in each elevation band is characterized by its water equivalent, $S_{\mathrm{sn},i}$, and its "cold content" indicated by a "temperature", $T_{\mathrm{sn},i}$. The function of this temperature is to delay the melting process whenever the temperature was very cold

before it climbs above zero. The two state variables, $S_{\mathrm{sn},i}$ and $T_{\mathrm{sn},i}$, fulfill the following differential equations:

$$\frac{\mathrm{d}S_{\mathrm{sn},i}}{\mathrm{d}t} = f_{\mathrm{snow},i}P_i - Q_{\mathrm{melt},i} \quad , \tag{A4}$$

$$\frac{\mathrm{d}T_{\mathrm{sn},i}}{\mathrm{d}t} = \frac{-\log(\theta_{\mathrm{G2}})}{U_t}\big(T_{\mathrm{mean},i} - T_{\mathrm{sn},i}\big) \quad . \tag{A5}$$

The amount of snow (in water units) is given by a simple mass balance between accumulation and melting. Temperature follows

the daily mean temperature with a rate constant of $-\log(\theta_{\mathrm{G2}})/U_t$ (note that $0 < \theta_{\mathrm{G2}} < 1$ and thus $\log(\theta_{\mathrm{G2}})$ is negative; see below for the justification of this parameterization). The snow melting rate is given by

$$Q_{\mathrm{melt},i} = \frac{\theta_{\mathrm{G1}}}{U_t} \cdot f_{\mathrm{SI}}(T_{\mathrm{mean},i} - T_{\mathrm{sm,th}}, \Delta T_{\mathrm{sm}}) \cdot f_{\mathrm{SH}}(T_{\mathrm{sn},i}, \Delta T_{\mathrm{sm}})$$

$$\cdot \left(1 - \exp\left(-\frac{S_{\mathrm{sn},i}}{S_{\mathrm{sn,th}}}\right)\right) \quad , \tag{A6}$$

which approaches the proportionality with temperature above the snow melt temperature $T_{\mathrm{mean},i} - T_{\mathrm{sm,th}}$ ($f_{\mathrm{SI}}$), and if the snow temperature, $T_{\mathrm{sn},i}$, is above zero ($f_{\mathrm{SH}}$), and if there still is snow present (last exponential term). These conditions are formulated by smooth transitions based on the equations (A2), (A1), and the exponential term.

Note that the analytical solution of equation (A5) under constant driving forces ($T_{\mathrm{mean},i} > T_{\mathrm{sm,th}}$) and disregarding the

smoothing of the transitions is given by

$$T_{\mathrm{sn},i}(t) = T_{\mathrm{mean},i} + \big(T_{\mathrm{sn},i}(0) - T_{\mathrm{mean},i}\big)\exp\left(\frac{\log(\theta_{\mathrm{G2}})}{U_t}t\right) \quad .$$

After one day ($U_t$) we thus get

$$T_{\mathrm{sn},i}(U_t) = T_{\mathrm{mean},i} + \big(T_{\mathrm{sn},i}(0) - T_{\mathrm{mean},i}\big)\theta_{\mathrm{G2}} = \theta_{\mathrm{G2}}T_{\mathrm{sn},i}(0) + (1 - \theta_{\mathrm{G2}})T_{\mathrm{mean},i} \quad .$$

which corresponds to the original time-discrete formulation Valery et al. (2014) and thus justifies our continuous-time approach.


Similarly to our justification for equation (A5), if $S_{\mathrm{sn},i} >> S_{\mathrm{sn,th}}$, we can neglect the exponential term in equation (A6) and if we further neglect smooting, integration over one day ($U_t$) leads to an integrated flux of $\theta_{\mathrm{G1}}(T_{\mathrm{mean},i} - T_{\mathrm{sm,th}})$ whereas for $S_{\mathrm{sn},i} << S_{\mathrm{sn,th}}$ we get approximately $\theta_{\mathrm{G1}}(T_{\mathrm{mean},i} - T_{\mathrm{sm,th}})S_{\mathrm{sn},i}/S_{\mathrm{sn,th}}$ as given by the discrete-time model Valery et al. (2014). This makes our model and the meaning of the parameters similar, but not identical to the Cemaneige model.


Finally, the input to the hydrologic model (per unit area) is given by the sum of the precipitation fractions falling as rain plus the sum of water from melting snow weighted by the relative areas of the elevation bands:

$$P_{\mathrm{eff}} = \sum_{i=1}^{n_{\mathrm{b}}} \frac{A_i}{A}\big((1 - f_{\mathrm{snow},i})P_i + Q_{\mathrm{melt},i}\big) \quad . \tag{A7}$$

Here, $A_i$ is the area of the elevation band $i$ and $A$ is the total area of the catchment.

This continuous-time snow model is now coupled with the published continuous-time version of the GR4 model Santos et al. (2018) given by the water balance differential equations for the two reservoirs S ($S_S$) and R ($S_R$) and the cascade ($S_{r,i}$):

$$\frac{\mathrm{d}S_S}{\mathrm{d}t} = P_s - E_s - Q_{\mathrm{perc}} \quad , \tag{A8}$$

$$\frac{\mathrm{d}S_{r,i}}{\mathrm{d}t} = \begin{cases} P_n - P_s + Q_{\mathrm{perc}} - \frac{n_r}{x_4} S_{r,i} & \text{for } i = 1 \\ \frac{n_r}{x_4}(S_{r,i-1} - S_{r,i}) & \text{for } i = 2, ..., n_r \end{cases} \quad , \tag{A9}$$

$$\frac{\mathrm{d}S_R}{\mathrm{d}t} = Q_9 + Q_F - Q_r \quad . \tag{A10}$$

The water fluxes in these equations are given by Santos et al. (2018)

$$P_n = \begin{cases} P_{\mathrm{eff}} - E_{\mathrm{pot}} & \text{for } P_{\mathrm{eff}} > E_{\mathrm{pot}} \\ 0 & \text{for } P_{\mathrm{eff}} \leq E_{\mathrm{pot}} \end{cases} \quad , \tag{A11}$$

$$E_n = \begin{cases} 0 & \text{for } P_{\mathrm{eff}} > E_{\mathrm{pot}} \\ E_{\mathrm{pot}} - P_{\mathrm{eff}} & \text{for } P_{\mathrm{eff}} \leq E_{\mathrm{pot}} \end{cases} \quad , \tag{A12}$$

$$P_s = P_n \left( 1 - \left( \frac{S_S}{x_1} \right)^\alpha \right) \quad , \tag{A13}$$

 $$E_s = E_n \left( 1 - \left( 1 - \frac{S_S}{x_1} \right)^\alpha \right) \quad . \tag{A14}$$

Note that we modified the equation

$$E_s = E_n \left( 2\frac{S_S}{x_1} - \left( \frac{S_S}{x_1} \right)^\alpha \right)$$

$$Q_{\mathrm{perc}} = \frac{x_1^{1-\beta}}{(\beta - 1)U_t} \nu^{\beta-1} S_S^\beta \quad , \tag{A15}$$

$$Q_{\mathrm{uh}} = \frac{n_{\mathrm{res}}}{x_4} S_{r,n_{\mathrm{res}}} \quad , \tag{A16}$$

$$Q_9 = \Phi Q_{\mathrm{uh}} \quad , \tag{A17}$$

$$Q_1 = (1 - \Phi)Q_{\mathrm{uh}} \quad , \tag{A18}$$

$$Q_{\mathrm{F}} = \frac{x_2}{x_3^{\omega}} S_{\mathrm{R}}^{\omega} \quad , \tag{A19}$$

$$Q_{\mathrm{R}} = \frac{x_3^{1-\gamma}}{(\gamma - 1)U_t} S_{\mathrm{R}}^{\gamma} \quad , \tag{A20}$$

$$Q = Q_{\mathrm{R}} + \max\bigl(0, Q_1 + Q_{\mathrm{F}}\bigr) \quad . \tag{A21}$$

Note that $x_2$ characterizes groundwater in- or output fed by or discharging to neighboring catchments. Set $x_2 = 0$ if you want to conserve mass within the catchment.

The parameters of the GR4neige model are listed together with their default values and ranges in Table A1.

**A3   HBV**

The HBV model is probably the most frequently used conceptual hydrologic model Bergström (1992); Lindström et al. (1997); Seibert (1999); Seibert and Vis (2012). As we use continous-time models in this paper, we develop a continuous-time model that is very similar to the original discrete-time HBV model. Figure A4 gives a schematic overview of the model.

We again distinguish $n_{\mathrm{b}}$ elevation bands to model snow cover. In contrast to the Cemaneige model, also the soil is resolved into these elevation bands. Within each elevation band, three state variables are used: snow, snow water (water content of the snowpack) and soil moisture.

We start with the same equation as for the GR4neige model to calculate the fraction of precipitation that falls as snow in
each elevation band, $i$:

$$f_{\mathrm{snow},i} = \begin{cases} 0 & \text{for } T_{\min,i} \geq T_{\mathrm{sf,th}} \\ \dfrac{T_{\mathrm{sf,th}} - T_{\min,i}}{T_{\max,i} - T_{\min,i}} & \text{for } T_{\max,i} > T_{\mathrm{sf,th}} \text{ and } T_{\min,i} < T_{\mathrm{sf,th}} \\ 1 & \text{for } T_{\max,i} \leq T_{\mathrm{sf,th}} \end{cases} \quad . \tag{A22}$$

| parameter | meaning | unit | default value | range* |
|---|---|---|---|---|
| $x_1$ | maximum capacity of production store | mm | 350 | $(0, \infty)$ |
| $x_2$ | intercatchment exchange (inflow) coeff. | mm/d | 0 | $(-\infty, \infty)$ |
| $x_3$ | capacity parameter of routing store | mm | 90 | $(0, \infty)$ |
| $x_4$ | base time of routing cascade | d | 1.7 | $(0, \infty)$ |
| $\theta_{G1}$ | maximum melting rate per degree above threshold | mm/d/°C | 3 | $(0, \infty)$ |
| $\theta_{G2}$ | cold capacity delay coefficient | - | 0.5 | $[0, 1]$ |
| $T_{sf,th}$ | threshold temperature for snowfall | °C | 0 | $(-\infty, \infty)$ |
| $T_{sm,th}$ | threshold temperature for snowmelt | °C | 0 | $(-\infty, \infty)$ |
| $\alpha$ | production store exponent | - | 2 | $(1, \infty)$ |
| $\beta$ | percolation exponent | - | 5 | $(1, \infty)$ |
| $\gamma$ | routing store outflow exponent | - | 5 | $(1, \infty)$ |
| $\Delta T_{sm}$ | temperature interval for snowmelt initiation | °C | 1 | |
| $S_{sn,th}$ | threshold snow level for turning off snowmelt | mm | 1 | |
| $\omega$ | intercatchment exchange exponent | - | 3.5 | |
| $\Phi$ | partion coefficient routing/outflow | - | 0.9 | |
| $\nu$ | percolation coefficient | - | 4/9 | |
| $n_b$ | number of elevation bands | - | 5 | |
| $n_r$ | number of routing cascade reservoirs | - | 11 | |

**Table A1.** Parameters of the GR4neige model. The upper part of the table lists the parameters that are always estimated for individual catchment fits, the middle part optional parameters to be added to the set of estimated parameters, and the lower part of the table lists parameters that are kept constant for these fits. * To avoid integration problems, the ranges are more strongly constrained during optimization.

The mass balance of snow is then described by the following equation:

$$\frac{dS_{sn,i}}{dt} = c_{sf} f_{snow,i} P_i - Q_{melt,i} + Q_{refr,i} \quad . \tag{A23}$$

Here, $c_{sf}$ is a parameter to empirically account for errors in snow measurement and evaporation of snow. In addition to the melting flow, $Q_{melt}$, the HBV model considers refreezing of snow water, $Q_{refr}$. The melting water flow is described similarly to the GR4neige model, except that there is no cold-content or snow temperature considered:

$$Q_{melt,i} = c_{melt} \cdot f_{SI}(T_{mean,i} - T_{sm,th}, \Delta T_{sm}) \cdot \left(1 - \exp\left(-\frac{S_{sn,i}}{S_{sn,th}}\right)\right) \quad . \tag{A24}$$

Refreezing is described similarly with the reverse temperature dependence and with a parameter $c_{fr}$ that reduces the rate compared to melting:

$$Q_{refr,i} = c_{fr} \, c_{melt} \cdot f_{SI}(T_{sm,th} - T_{mean,i}, \Delta T_{sm}) \cdot \left(1 - \exp\left(-\frac{S_{sw,i}}{S_{sw,th}}\right)\right) \quad . \tag{A25}$$

The total water flow production in each elevation band is given by the sum of melting snow and precipitation that falls as rain, $Q_{melt,i} + (1 - f_{snow,i}) P_i$. Only a fraction of this water flow feeds the snow water reservoir, as this flux is limited by the amount

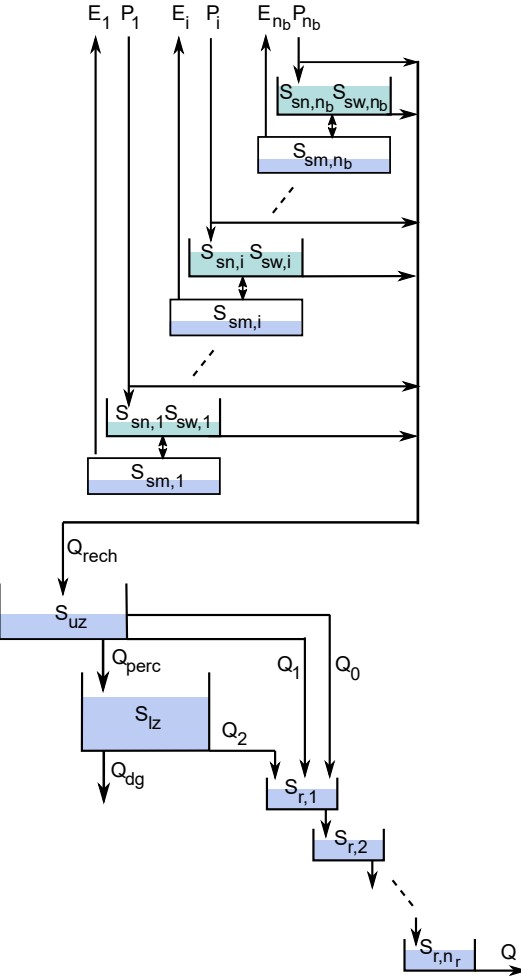

**Figure A4.** Schematic diagram of the HBV model as used in this paper.

of snow and by approaching the water holding capacicty of the snowpack, $c_{\mathrm{wh}}$:

$$
Q_{\mathrm{sw},i} = \begin{cases} 0 & \text{if } S_{\mathrm{sw},i} \geq c_{\mathrm{wh}} S_{\mathrm{sn}} \\[2em] \begin{aligned} &\bigl(Q_{\mathrm{melt},i} + (1 - f_{\mathrm{snow},i}) P_i\bigr) \\ &\cdot \left(1 - \exp\left(-\frac{S_{\mathrm{sn},i}}{S_{\mathrm{sn,th}}}\right)\right) \\ &\cdot \left(1 - \exp\left(-\frac{c_{\mathrm{wh}} S_{\mathrm{sn},i} - S_{\mathrm{sw},i}}{S_{\mathrm{sw,th}}}\right)\right) \end{aligned} & \text{if } S_{\mathrm{sw},i} < c_{\mathrm{wh}} S_{\mathrm{sn}} \end{cases} \tag{A26}
$$

The remaining part $Q_{\mathrm{melt},i} + (1 - f_{\mathrm{snow},i}) P_i - Q_{\mathrm{sw},i}$, together with snow water release, $Q_{\mathrm{rel},i}$, leaves the snowpack:

$$
Q_{\mathrm{sn},i} = Q_{\mathrm{melt},i} + (1 - f_{\mathrm{snow},i}) P_i - Q_{\mathrm{sw},i} + Q_{\mathrm{rel},i} \quad . \tag{A27}
$$

Snow water release is the most challenging part of the continuous-time formulation of the HBV model. It is needed as the relative water content would increase beyond the water colding capacity of the snowpack, $c_{\mathrm{wh}}$, when snow melts even in the absence of feeding water. In the original HBV model, excess water beyond the water holding capacity is just discharged at
each time step. To avoid a discontinuous flux, we accept a deviation from the discrete-time model by allowing for an increasing release of snow water already below the water holding capacity, $c_{\mathrm{wh}}$:

$$Q_{\mathrm{rel},i} = \begin{cases} 0 & \text{if } S_{\mathrm{sn},i} = 0 \\ \dfrac{S_{\mathrm{sw},i}}{c_{\mathrm{wh}} S_{\mathrm{sn},i}} Q_{\mathrm{melt},i} & \text{if } S_{\mathrm{sn},i} > 0 \end{cases} \quad . \tag{A28}$$

This finally leads to the differential equation for snow water:

$$\frac{\mathrm{d}S_{\mathrm{sw},i}}{\mathrm{d}t} = Q_{\mathrm{sw},i} - Q_{\mathrm{refr},i} - Q_{\mathrm{rel},i} \quad . \tag{A29}$$

The water leaving the snowpack, $Q_{\mathrm{sn},i}$, is now divided into a fraction that feeds soil moisture and a fraction that recharges groundwater with the original nonlinear relationship with exponent $\beta$ as in the original HBV model:

$$\frac{\mathrm{d}S_{\mathrm{sm},i}}{\mathrm{d}t} = Q_{\mathrm{sn},i}\left(1 - \left(\frac{S_{\mathrm{sm},i}}{S_{\mathrm{fc}}}\right)^{\beta}\right) - E_{\mathrm{pot}}\left(1 - \exp\left(-\frac{S_{\mathrm{sm}}}{S_{\mathrm{sm,th}}}\right)\right)\exp\left(-\frac{S_{\mathrm{sn}}}{S_{\mathrm{sn,th}}}\right) , \tag{A30}$$

$$Q_{\mathrm{rech}} = \sum_{i=1}^{n_{\mathrm{b}}} \frac{A_i}{A} Q_{\mathrm{sn},i}\left(\frac{S_{\mathrm{sm},i}}{S_{\mathrm{fc}}}\right)^{\beta} \quad . \tag{A31}$$

Groundwater is then described by water content of an upper zone, $S_{\mathrm{uz}}$, and a lower zone, $S_{\mathrm{lz}}$

$$\frac{\mathrm{d}S_{\mathrm{uz}}}{\mathrm{d}t} = Q_{\mathrm{rech}} - Q_{\mathrm{perc}} - Q_0 - Q_1 \quad , \tag{A32}$$

$$\frac{\mathrm{d}S_{\mathrm{lz}}}{\mathrm{d}t} = Q_{\mathrm{perc}} - Q_2 - Q_{\mathrm{dg}} \quad , \tag{A33}$$

with a percolation flux from the upper to the lower zone given by

$$Q_{\mathrm{perc}} = c_{\mathrm{perc}}\left(1 - \exp\left(-\frac{S_{\mathrm{uz},i}}{S_{\mathrm{uz,th}}}\right)\right) \quad , \tag{A34}$$

and outfluxes given by

$$Q_0 = k_0 f_{\mathrm{SI}}(S_{\mathrm{uz}} - S_{\mathrm{uz,div}}, S_{\mathrm{uz.th}}) \quad , \tag{A35}$$

$$Q_1 = k_1 S_{\mathrm{uz}} \quad , \tag{A36}$$

$$Q_2 = k_2 S_{\mathrm{lz}} \quad , \tag{A37}$$

$$Q_{\mathrm{dg}} = k_{\mathrm{dg}} S_{\mathrm{lz}} \quad . \tag{A38}$$

These process formulations follow exactly the HBV model with the single exception of the additional flow to deep groundwater, $Q_{\mathrm{dg}}$, that, if $k_{\mathrm{dg}}$ is negative, can also describe a feed from neighboring catchments. It turned out that some higher catchments need such a term that is similar (except from the sign) to the term characterized by the parameter $x_2$ of the GR4snow model.

The final model component is the a reservoir cascade that describes flow routing:

$$\frac{\mathrm{d}S_{\mathrm{r},i}}{\mathrm{d}t} = \begin{cases} Q_0 + Q_1 + Q_2 - n_{\mathrm{r}} k_{\mathrm{r}} S_{\mathrm{r},1} & \text{for } i = 1 \\ n_{\mathrm{r}} k_{\mathrm{r}} (S_{\mathrm{r},i-1} - S_{\mathrm{r},i}) & \text{for } i = 2, ..., n_{\mathrm{r}} \end{cases} . \tag{A39}$$

The catchment outflow is then given as the outflow from the final routing reservoir:

$$Q = n_{\mathrm{r}} k_{\mathrm{r}} S_{\mathrm{r},n_{\mathrm{r}}} \quad . \tag{A40}$$

The parameters of this continuous-time version of the HBV model are listed together with their default values and ranges in Table A2.

| parameter | meaning | unit | default value | range* |
|---|---|---|---|---|
| $c_{\mathrm{melt}}$ | maximum melting rate per degree above threshold | mm/(°C d) | 3 | $(0, \infty)$ |
| $S_{\mathrm{fc}}$ | maximum soil moisture level | mm | 100 | $(0, \infty)$ |
| $S_{\mathrm{uz,div}}$ | division betw. lower and upper part of upper groundw. | mm | 10 | $(0, \infty)$ |
| $c_{\mathrm{perc}}$ | maximum percolation water flow | mm/d | 1.5 | $(0, \infty)$ |
| $k_0$ | water release coeff. from upper part of upper groundw. | 1/d | 1.5 | $(0, \infty)$ |
| $k_1$ | water release coefficient from upper groundwater zone | 1/d | 0.1 | $(0, \infty)$ |
| $k_2$ | water release coefficient from lower groundwater zone | 1/d | 0.1 | $(0, \infty)$ |
| $k_r$ | water release coefficient of routing cascade | 1/d | 2 | $(0, \infty)$ |
| $k_{\mathrm{dg}}$ | rate coeff. for outflow from gw. to deep gw. | 1/d | 0 | $(-\infty, \infty)$ |
| $\beta$ | exponent for water distribution to soil and groundwater | - | 3 | $(2, \infty)$ |
| $T_{\mathrm{sf,th}}$ | threshold temperature for snowfall | °C | 0 | $(-\infty, \infty)$ |
| $T_{\mathrm{sm,th}}$ | threshold temperature for snowmelt | °C | 0 | $(-\infty, \infty)$ |
| $\Delta T_{\mathrm{sm}}$ | temperature interval for snowmelt initiation | °C | 1 | |
| $S_{\mathrm{sn,th}}$ | threshold snow level for turning off snowmelt | mm | 1 | |
| $S_{\mathrm{sw,th}}$ | threshold snow water level for turning off refreezing | mm | 0.2 | |
| $S_{\mathrm{sm,th}}$ | threshold water level of unsat. zone for turning off evap. | mm | 0.5 | |
| $S_{\mathrm{uz,th}}$ | threshold for turning off percol. from upper groundw. | mm | 1 | |
| $c_{\mathrm{sf}}$ | snowfall correction factor | - | 1 | |
| $c_{\mathrm{fr}}$ | coefficient of reduction of freezing rel. to melting rate | - | 1 | |
| $c_{\mathrm{wh}}$ | water holding fraction in snowpack | - | 0.1 | |
| $c_{\mathrm{e}}$ | multiplication factor for potential evaporation | - | 1 | |
| $n_{\mathrm{b}}$ | number of elevation bands | - | 5 | |
| $n_{\mathrm{r}}$ | number of routing cascade reservoirs | - | 5 | |

**Table A2.** Parameters of the HBV model. The upper part of the table lists the parameters that are always estimated for individual catchment fits, the middle part optional parameters to be added to the set of estimated parameters, and the lower part of the table lists parameters that are kept constant for these fits. * To avoid integration problems, the ranges are more strongly constrained during optimization.

## Appendix B: LSTM

The LSTM architecture was already successfully tested for predictions of streamflow Feng et al. (2020, 2021); Ma et al. (2021), soil moisture Fang et al. (2017, 2019, 2020), stream temperature Rahmani et al. (2021), snow water equivalent Song et al. (2024); Cui et al. (2023), lake water temperature Read et al. (2019), dissolved oxygen Zhi et al. (2023) and nitrate Saha et al. (2023). LSTM is a type of Recurrent Neural Network (RNN) that learns from sequential data. The difference from a simple RNN is that LSTM has "memory states" and "gates", which allow it to learn how long to retain the state information, what to forget, and what to output. The forward pass of the LSTM model is described by the following equations:

$$\text{input transformation: } \mathbf{x}^t = \text{ReLU}\left(\mathbf{W_I}\mathbf{I}^t + \mathbf{b_I}\right) \tag{B1}$$

$$\text{input node: } \mathbf{g}^t = \tanh\left(\mathcal{D}\left(\mathbf{W_{gx}}\mathbf{x}^t\right) + \mathbf{b_{gx}} + \mathcal{D}\left(\mathbf{W_{gh}}\mathbf{h}^{t-1}\right) + \mathbf{b_{gh}}\right) \tag{B2}$$

$$\text{input gate: } \mathbf{i}^t = \sigma\left(\mathcal{D}\left(\mathbf{W_{ix}}\mathbf{x}^t\right) + \mathbf{b_{ix}} + \mathcal{D}\left(\mathbf{W_{ih}}\mathbf{h}^{t-1}\right) + \mathbf{b_{ih}}\right) \tag{B3}$$

$$\text{forget gate: } \mathbf{f}^t = \sigma\left(\mathcal{D}\left(\mathbf{W_{fx}}\mathbf{x}^t\right) + \mathbf{b_{fx}} + \mathcal{D}\left(\mathbf{W_{fh}}\mathbf{h}^{t-1}\right) + \mathbf{b_{fh}}\right) \tag{B4}$$

$$\text{output gate: } \mathbf{o}^t = \sigma\left(\mathcal{D}\left(\mathbf{W_{ox}}\mathbf{x}^t\right) + \mathbf{b_{ox}} + \mathcal{D}\left(\mathbf{W_{oh}}\mathbf{h}^{t-1}\right) + \mathbf{b_{oh}}\right) \tag{B5}$$

$$\text{cell state: } \mathbf{s}^t = \mathbf{g}^t \odot \mathbf{i}^t + \mathbf{s}^{t-1} \odot \mathbf{f}^t \tag{B6}$$

$$\text{hidden state: } \mathbf{h}^t = \tanh\left(\mathbf{s}^t\right) \odot \mathbf{o}^t \tag{B7}$$

$$\text{output: } \mathbf{y}^t = \mathbf{W_{hy}}\mathbf{h}^t + \mathbf{b_y} \tag{B8}$$

where $\mathbf{I}^t$ represents the raw inputs for the time step, ReLU is the rectified linear unit, $\mathbf{x}^t$ is the vector to the LSTM cell, $\mathcal{D}$ is the dropout operator, $\mathbf{W}$'s are network weights, $\mathbf{b}$'s are bias parameters, $\sigma$ is the sigmoidal function, $\odot$ is the element-wise multiplication operator, $\mathbf{g}^t$ is the output of the input node, $\mathbf{i}^t$, $\mathbf{f}^t$, $\mathbf{o}^t$ are the input, forget, and output gates, respectively, $\mathbf{h}^t$ represents the hidden states, $\mathbf{s}^t$ represents the memory cell states, and $\mathbf{y}^t$ is the predicted output.

The LSTM was calibrated using the catchment attributes shown in Table B1 Addor et al. (2017).

| Attributes | Reduced Set | Description | Unit |
|---|---|---|---|
| elev_mean | | Catchment mean elevation | m |
| slope_mean | x | Catchment mean slope | m/km |
| area_gages2 | x | Catchment area (GAGESII estimate) | km$^2$ |
| high_prec_freq | | Frequency of high precipitation days | days/year |
| high_prec_dur | | Average duration of high precipitation | days |
| low_prec_freq | | Frequency of dry days | days/year |
| low_prec_dur | | Average duration of dry periods | days |
| frac_forest | x | Forest fraction | - |
| lai_max | x | Maximum monthly mean of the leaf area index | - |
| lai_diff | x | Difference between max. and min. monthly mean leaf area index | - |
| dom_land_cover_frac | x | Fraction of catchment area associated with dominant land cover | - |
| dom_land_cover | x | Dominant land cover type | - |
| root_depth_50 | x | Root depth at 50th percentiles | m |
| soil_depth_statsgo | x | Soil depth | m |
| soil_porosity | x | Volumetric soil porosity | - |
| soil_conductivity | x | Saturated hydraulic conductivity | cm/hr |
| max_water_content | x | Maximum water content | m |
| geol_1st_class | x | Most common geologic class in the catchment | - |
| geol_2nd_class | x | Second most common geologic class in the catchment | - |
| geol_porostiy | x | Subsurface porosity | - |
| geol_permeability | x | Subsurface permeability | m$^2$ |
| p_mean | | Mean daily precipitation | mm/day |
| pet_mean | | Mean daily PET | mm/day |
| p_seasonality | | Seasonality and timing of precipitation | - |
| frac_snow | | Fraction of precipitation falling as snow | - |
| aridity | | PET/P | - |

**Table B1.** Full and reduced sets of catchment attributes used for the calibration of the LSTM models Addor et al. (2017).

*Code availability.* This work is based on published and publicly available data sets and our code is publicly available (Conceptual models: https://doi.org/10.25678/000CQ0. LSTM: http://doi.org/10.5281/zenodo.3993880).

*Author contributions.* PR developed the concept of the paper, implemented the conceptual hydrologic models, did all simulations with these models and wrote the first version of the paper. KM trained the LSTMs and ran all the simulations with these models. All co-authors
700   contributed to stimulating discussions about the paper concept and the results and to the revision of the first paper version and the finalization of the paper.

*Competing interests.* At least one of the (co-)authors is a member of the editorial board of *Hydrology and Earth System Sciences*.

*Acknowledgements.* We thank Jan Seibert for clarifications regarding the (discrete-time) HBV model. CS thanks Eawag's sabbatical support that enabled this collaboration. Finally, we thank Scott Steinschneider and two anonymous reviewers for their comments and suggestions to
705   improve the manuscript.

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
