# Peer review of "Metamorphic Testing of Machine Learning and Conceptual Hydrologic Models"

_Hydrology and Earth System Sciences, 2023_

## Author Comment (AC1)

**CC1**

I read this paper with interest, as I agree that such metamorphic tests on ML hydrologic models are needed to assess their appropriateness for certain hydrologic modeling applications like projections under climate change.

My main comment is that I think the authors could contextualize their study with past work that has conducted a similar exploration. The first paper that I am aware of which attempted a metamorphic test on an LSTM was Razavi (2021) (see their Figure 11). They only considered an LSTM fit to one site, and so there are limitations to that work, but I think it is important to recognize it. Afterwards, Wi and Steinschneider (2022) conducted a similar metamorphic test as conducted in the present study, using both 1) an LSTM and physics-informed LSTMs fit to 15 sites across California, as well as an LSTM fit across the entire CAMELS dataset. They found related challenges with LSTM projections under warming as found in this work.

Therefore, I recommend that the authors adjust their Introduction to recognize these past studies, and then to articulate how their work provides a contribution over these past studies. I believe this is very straightforward, as the present study 1) considers changes in precipitation as well; 2) explores responses separately by basin elevation and temperature; and 3) explore sensitivity to calibration choices (this later one was particularly helpful to see). In addition, I might adjust the Summary and Conclusion to discuss the results of the present study in comparison to the metamorphic results seen in Wi and Steinschneider (2022), in order to help synthesize related results in the literature.

References

Razavi, S. (2021). Deep learning, explained: Fundamentals, explainability, and bridgeability to process-based modelling, Environmental Modelling and Software, 105159, https://doi.org/10.1016/j.envsoft.2021.105159.

Wi, S., & Steinschneider, S. (2022). Assessing the physical realism of deep learning hydrologic model projections under climate change. Water Resources Research, 58, e2022WR032123. https://doi.org/10.1029/2022WR032123

We thank the reviewer for pointing our attention to these papers. We fully agree with adapting the introduction and conclusions in the sense suggested by the reviewer.

These papers were also useful for identifying newer papers that cite these papers and are useful for additional references regarding hybrid models in the discussion:

Ng K.W., Huang Y.F., Koo C.H., Chong K.L., El-Shafie A., Ahmed, A.N. A review of hybrid deep learning applications for streamflow forecasting. Journal of Hydrology 625 (2023) 130141.

Razavi, S., Hannah, D.M., Elshorbagy, A., Kumar, S. Marshall, L., Solomatine, D.P., Dezfuli, A., Sadegh, M. and Famiglietti, J. (2022) Coevolution of machine learning and process-based modelling to revolutionize Earth and environmental sciences: A perspective. Hydrological Processes. 2022;36:e14596.

Zhong, L., Lei, H., & Gao, B. (2023). Developing a physics-informed deep learning model to simulate runoff response to climate change in Alpine catchments. Water Resources Research, 59, e2022WR034118.

---

## Author Comment (AC2)

**RC2:**

I believe this is a significant paper, I am looking forward to its publishing, however, I do believe some improvements can and should be made first. The paper lacks a bit of specificity when it comes to implementation details of the metamorphic testing approach. In particular I find the choice of model types in the multi-model approach important, the merits of which are not particularly covered in the paper until the conclusion section. I would suggest this be done in a more sophisticated way around lines 88-89. Other considerations should also be specified and guidance provided as to what correct versus incorrect implementation entails, it is at the moment a bit too broad on this front.

> We agree to add a few sentences about the considered model structures and the underlying motivation. In particular that we can expect physically realistic responses from conceptual models as long as the modified inputs do not lead to changes in catchment properties, such as vegetation and soil structure, whereas the machine learning models bear the potential of covering such changes, but their accuracy/correctness beyond the calibration data is difficult to estimate.

Please see more details below:

Line 19: Odd source placement.

> The source is primarily about the hydrological data and catchment properties, not about all the research stimulated by that. For this reason, moving the citation to the end of the sentence would not be correct. We will add examples of the research stimulated by these data at the end of the sentence and hope that this will clarify both citation placements.

Line 24 and 30 please specify whether you are talking about deep learning, and if you do then clarify right away. Additionally, at its current location the Shen (2018) citation seems out of place.

> We intentionally wanted to be general here. We will add a sentence that these achievements were primarily based on deep learning methodologies and even more specifically on LSTM models. We will remove the Shen (2018) citation here.

Line 27: Good that you point this out, please also provide some examples.

> We will provide some references to the combination of both points (i) and (ii). Some of them are already cited in line 22 above.

Lines 33 to 36: Seems out of place, maybe move or integrate into discussion meaningfully.

> We will move this sentence further up after mentioning the success of the machine-learning (deep learning, LSTM) approaches (after the citation of Ma et al. 2021) and mention the availability of more data sets later.

Lines 82 and 85-88: I am glad you point these out!

> We agree that it is very important to be aware of the limitations of this approach also.

Line 88-89: I was hoping for more detail.

> Since the models are described in section 2.2 and extensively in the appendices A and B, and the calibration options were partly motivated by the results (and thus described in section 3.3, in particular in Table 1), we tried to avoid repetitions. In the new version, we will reference these sections and appendices so that the reader can easily find where to search for more details.

Line 137: Re-cite in the bracket

> We agree.

Line 150: Reduce the overall size of this paragraph and the one around line 130 by focusing only on the basins that will appear in the study. Perhaps a more concise mention near Line 170 and 175 of the expected effects would be best. A schematic can also be useful in showing the expected effects for the basins in your study.

We agree and suggest to add the following figure of the expected qualitative changes in the response at the catchment outlet and shorten the text accordingly.

[Figure]

The expected qualitative changes to the precipitation increase are indicated in blue, those to the temperature increase in red. The qualitative changes are expected to be less smooth than in these figures that just represent general trends due to shorter-term precipitation and temperature fluctuations.

Line 195: Please specify the choice of optimizer.

We will add a sentence to indicate that the chosen optimizer for our model is AdaDelta (lr=1.0, rho=0.9).

Line 220: Remain consistent throughout with your word-choice of training versus calibration. (However, you should keep the statement equating these terms for researchers from different backgrounds that may be used to one or the other.)

We agree. It should have been "training" here. We will explain that we use "training" for machine learning models and "calibration" for the conceptual models as it is typical in the corresponding literature, but that both are conceptually the same although often different algorithms are used.

Figure 2: Utilize symbols for black and white version distinguishability of basin classification.

We will adapt the symbols accordingly.

Figure 3/4/5: Particularly the top 2 panels are difficult to read in black and white, double check with the editor whether this would need to be adjusted.

We will adapt the color palette for color-blind readers.

Line 323-324: Perhaps use a different loss function with guaranteed convexity to avoid this problem. You can still use NSE for evaluation of course to stay consistent with the other models. (I would recommend you do this at least for the reanalysis)

We agree that experimenting with different loss functions is important but this is not a major focus of this paper as we wanted to conform with common practice with LSTM modelling in hydrology, which so far was mainly based on NSE as a loss function.

We also do not see that we could avoid the existence of multiple (many!) local minima of an LSTM model by changing the loss function as the existence of multiple minima originates from the complex model structure and not from the loss function applied to the residuals between model and data. Using different seeds at least partly demonstrates the variability that we can expect from ending up in different local minima.

Line 363: Instead of "was not investigated" saying "this was not the main aim" would be better.

We agree.

Line 412-414: Make these claims and future research directions more specific. Why ML and which research will be necessary to achieve this?

We will mention here, that adding catchment properties, such as vegetation and soil, to conceptual or physical hydrological models, could in principle improve the predictive properties of these models under strong input changes. However, the parameterization of the processes for vegetation and soil structures is very difficult and can lead to higher model structure uncertainty. For this reason, a ML approach that is trained by a large variety of catchments under different climatic conditions that lead to different catchment properties may be a promising approach. However, as seen in the current study, this potential is difficult to realize also and a hybrid approach could be promising.

Last paragraph: In the conclusion new ideas should not be first mentioned as it is done here. Citations should also not be in the conclusion section for this reason. Please mention these points earlier in the discussion and just summarize them in the conclusion.

We will discuss these points already in the discussion and move the citations there.

Thank you for your submission! I am looking forward to reviewing the revised version of this manuscript!

---

## Author Comment (AC3)

**RC1:**

Reichert and co-authors describe a study in which they test the qualitative response (metamorphic testing) of two conceptual hydrologic models and a deep learning model trained on CAMELS-US dataset to perturbed temperature and precipitation, aimed at mimicking the qualitative performance of these models under climate change scenarios. The deep learning model (LSTM) outperforms the conceptual models during calibration and validation, but exhibited unexpected hydrologic response in low-elevation basins when temperature was perturbed. Solely training on the low-elevation basins from CAMELS-US improved this qualitative response to the perturbed temperature, suggesting that fine-tuning or limiting datasets to prediction task may help improve out-of-bounds predictions. I provide some comments below that I think will improve the manuscript.

- Generally, I encourage the authors to attempt to summarize all the basins used for metamorphic testing rather than providing individual plots across all basins in the supplementary. There does not appear to be a reason why certain basins are displayed in the main text vs. supplementary. Summarizing across all basins used will help the reader understand better if the pattern is common without having to look through dozens of individual plots in supplementary.

  We agree that a numerical summary across the basins improves the manuscript and we suggest to add the following figure with the summary statistics. As we cannot compare with the truth or with data, we compare the sensitivities of the different approaches by calculating the root mean squared differences of all pairs of approaches and average them over the seeds (if available) and over the catchments within the same class. We suggest to include the results for the most important case, the temperature sensitivities of the low altitude catchments in the paper:

[Figure]

  In this figure, we clearly see that the LSTM sensitivities approach those of GR4neige (decreasing differences [in mm/d]) when moving from the basic LSTM to the LSTM_211 (first column) and they also approach those of HBV (second column). In parallel to that, we see that the sensitivities of the LSTM deviate more and more from the basic LSTM when moving from the basic LSTM to the LSTM_211. These are the changes discussed in the paper.

  Still, we believe that showing one example for each class of catchments in the paper is very important to demonstrate the shapes of the sensitivities and, for being transparent, showing the results for all chosen catchments in the supporting information is useful to demonstrate that the response is typical for the class. For this reason, we suggest to add the summary table/figure but to keep the examples.

Line 40 and elsewhere: Instead of 'modified driving forces', could you use something more generic like 'out-of-bounds predictions'?

> We think that "modified driving forces" more clearly expresses what we do than "out-of-bound predictions".

- Line 114-115: the initial clause seems a bit clunky. Would it be clearer to either remove the first clause entirely or consider placing it as a separate sentence?

  > We agree and will split the sentence to express the point more clearly.

- Lines 120-139: this is a flat 10% increase in precipitation for every precipitation data point in the dataset? What if there is no rain on a given day? I assume that will still be 0 precipitation increase scenario given the equation 1 but clarification would be helpful.

  > This is correct, $1.1 \times 0 = 0$. We will add a sentence to clarify this.

- Line 160: can you give examples of what these precipitation or temperature related attributes would be?

  > The full and reduced sets of attributes are given in Table B1. We will add this reference in line 160. Examples are "mean daily precipitation" and "fraction of precipitation falling as snow".

- Line 168: what was the validation NSE range for these catchments?

  > All individual NSE values for both, calibration and validation periods, are given in the Figures in the Supporting Information. The range for calibration is 0.82-0.92, the range for validation is 0.67-0.91. We will add this to the paper.

- Line 213: I encourage the authors to include a link to the working repository at the moment, or a draft code release.

  > The link for the working repository for the conceptual models is https://gitlab.com/p.reichert/hyperflex.
  > The LSTM code used in this work can be accessed at http://doi.org/10.5281/zenodo.3993880.

- Figure 2: describe in the legend what the numbers next to the points represent.

  > These are the identifiers of the CAMELS basin. We will add this statement to the legend.

- Line 253: clarify that the sensitivities are in relation to the outlet discharge and not the overall model performance – 'sensitivities of the models are essentially negative' makes it sound like the models had a poor/unexpected outcome, but this is quite the opposite. I suggest changing to something like, "The predicted outlet discharge for the GR4neige and HBV models was lower with increased temperature, which is expected … "

  > $\Delta Q_P$ and $\Delta Q_T$ are defined by the equations (1) and (2), respectively. They are negative if the predicted output is lower than for the base simulation and positive if it is higher. We will add an explanation about the meaning of the sensitivities and what is expected.

- Figure 3: Please change the colors of the third panel to be different than the colors used to indicate the different types of models in the other panels? This is confusing to switch the meaning of the colors in the same figure.

  > We agree and will do that.

- Figure 3: for the top panel, why is there a y-axis that extends to -4 when there are no negative values? Also seems like the max differences are cutoff at the top of the y-axis, please extend higher.

  We think that it makes sense to have the same scale for the sensitivities across all catchments to make it easier to compare them. The chosen scale if a compromise of seeing sufficient detail and not cutting some of the sensitivities too much.

- Figure 3: for the fourth panel, indicate what the black circles are – I assume they are observations

  Yes, they are. We will add this statement to the legend.

- Figure 4: please include the full legend here so the reader doesn't have to refer to a separate figure

  We agree and will do that.

- Line 276: change "rainfall" to "precipitation" as a lot of this precipitation is falling as snow in this basin and other basins.

  We agree. Thank you for the hint.

- Figure 4: it is hard to distinguish the different model traces on here and I cannot tell what I'm supposed to take away from the bottom panel. I suggest splitting out the LSTM traces into a separate panel and/or show the deviation of the LSTM_x compared to the base LSTM results.

  We agree that this is difficult to see and will improve the signatures.

- Lines 395-402: I'm curious if the authors tried pre-training on the entire dataset with early stopping criteria as to not overfit, and then fine tune on the reduced dataset with only low-elevation basins. That seems like it might be the best of both worlds – providing better fits by using more data but also passing the metamorphic test.

  We agree that strategies like pre-training can be highly effective. In a separate study (Ma et al. 2021, cited in the paper), we have leveraged pre-training and transfer learning technique to enhance the performance on smaller datasets. However, for the purpose of this study, our focus is on employing a standard LSTM model. This approach is intended to provide insights into the typical methodologies used in training ML models and to highlight the discernible differences that emerge from such conventional training practices.

- Lines 398-399: by how much did the quality of fit deteriorate? By 1%, 30%? I suggest adding in a quantitative measure so the readers can evaluate how much the tradeoff is for passing the metamorphic test with less data.

  We agree. Taking the basin 11468500 as an example, when the number of data reduces from 211 to 99, the validation NSE drops from 0.86 to 0.30 which is far below the range of validation NSE values for all approaches used in the paper (0.67-0.91).

- Lines 412-414: See Topp et al. 2023 https://doi.org/10.1029/2022WR033880 for comparison of ML architectures to prediction in unseen conditions. They suggest ML process/physics guidance helps improve predictions in unseen conditions. Likewise, see Read et al. 2019 https://doi.org/10.1029/2019WR024922 for out-of-bounds predictions using process-guidance for ML models.

  Thank you very much for these hints. These papers are about stream water temperature which is not a topic of this paper. We can still add these references to the discussion that hybrid approaches are also relevant in other areas of hydrology and beyond.

---

## Author Response (AR1)

**CC1**

I read this paper with interest, as I agree that such metamorphic tests on ML hydrologic models are needed to assess their appropriateness for certain hydrologic modeling applications like projections under climate change.

My main comment is that I think the authors could contextualize their study with past work that has conducted a similar exploration. The first paper that I am aware of which attempted a metamorphic test on an LSTM was Razavi (2021) (see their Figure 11). They only considered an LSTM fit to one site, and so there are limitations to that work, but I think it is important to recognize it. Afterwards, Wi and Steinschneider (2022) conducted a similar metamorphic test as conducted in the present study, using both 1) an LSTM and physics-informed LSTMs fit to 15 sites across California, as well as an LSTM fit across the entire CAMELS dataset. They found related challenges with LSTM projections under warming as found in this work.

Therefore, I recommend that the authors adjust their Introduction to recognize these past studies, and then to articulate how their work provides a contribution over these past studies. I believe this is very straightforward, as the present study 1) considers changes in precipitation as well; 2) explores responses separately by basin elevation and temperature; and 3) explore sensitivity to calibration choices (this later one was particularly helpful to see). In addition, I might adjust the Summary and Conclusion to discuss the results of the present study in comparison to the metamorphic results seen in Wi and Steinschneider (2022), in order to help synthesize related results in the literature.

References

Razavi, S. (2021). Deep learning, explained: Fundamentals, explainability, and bridgeability to process-based modelling, Environmental Modelling and Software, 105159, https://doi.org/10.1016/j.envsoft.2021.105159.

Wi, S., & Steinschneider, S. (2022). Assessing the physical realism of deep learning hydrologic model projections under climate change. Water Resources Research, 58, e2022WR032123. https://doi.org/10.1029/2022WR032123

We thank the reviewer for pointing our attention to these papers. We fully agree and adapted the introduction as well as the discussion in the sense suggested by the reviewer.

These papers were also useful for identifying newer papers. We revised the discussion and extended the literature review.

**RC2:**

I believe this is a significant paper, I am looking forward to its publishing, however, I do believe some improvements can and should be made first. The paper lacks a bit of specificity when it comes to implementation details of the metamorphic testing approach. In particular I find the choice of model types in the multi-model approach important, the merits of which are not particularly covered in the paper until the conclusion section. I would suggest this be done in a more sophisticated way around lines 88-89. Other considerations should also be specified and guidance provided as to what correct versus incorrect implementation entails, it is at the moment a bit too broad on this front.

> We agree to add a few sentences about the considered model structures and the underlying motivation. In particular that we can expect physically realistic responses from conceptual models as long as the modified inputs do not lead to changes in catchment properties, such as vegetation and soil structure, whereas the machine learning models bear the potential of covering such changes, but their accuracy/correctness beyond the calibration data is difficult to estimate.

> We added a few sentences of these considerations and a link to the model description in section 2.2 and in the appendices A and B.

Please see more details below:

Line 19: Odd source placement.

> The source refers to hydrological data and catchment properties and not about all the research stimulated by that. For this reason, moving the citation to the end of the sentence would not be correct. We added examples of the research stimulated by these data at the end of the sentence.

Line 24 and 30 please specify whether you are talking about deep learning, and if you do then clarify right away. Additionally, at its current location the Shen (2018) citation seems out of place.

> We intentionally wanted to be general here. We added the statement that these models are typically based on deep learning architectures in the form of LSTM models. We removed the Shen (2018) citation here.

Line 27: Good that you point this out, please also provide some examples.

> We added some references.

Lines 33 to 36: Seems out of place, maybe move or integrate into discussion meaningfully.

> We moved this sentence further up after mentioning the success of the machine-learning (deep learning, LSTM) approaches (after the citation of Ma et al. 2021) and mention the availability of more data sets later.

Lines 82 and 85-88: I am glad you point these out!

> We agree that it is very important to be aware of the limitations of this approach also. We clearly state (in section 2.1) that this approach complements and does not at all replace any of the more commonly performed validation techniques. We also added this point to the abstract.

Line 88-89: I was hoping for more detail.

> We added a description of our motivation to include conceptual and machine learning models and a reference to section 2.2 and to the appendices A and B where the models are described in detail.

Line 137: Re-cite in the bracket

> Done.

Line 150: Reduce the overall size of this paragraph and the one around line 130 by focusing only on the basins that will appear in the study. Perhaps a more concise mention near Line 170 and 175 of the expected effects would be best. A schematic can also be useful in showing the expected effects for the basins in your study.

We agree and added the following new Figure 1 that shows the simplified expected changes in the response at the catchment outlet and we adapted and shortened the description of the expected changes.

[Figure]

The expected qualitative changes to the precipitation increase are indicated in blue, those to the temperature increase in red. The qualitative changes are expected to be less smooth than in these figures that just represent general trends due to shorter-term precipitation and temperature fluctuations.

Line 195: Please specify the choice of optimizer.

We added the information about the chosen optimizers for the LSTM (AdaDelta) and the conceptual models (LBFGS) with the corresponding references.

Line 220: Remain consistent throughout with your word-choice of training versus calibration. (However, you should keep the statement equating these terms for researchers from different backgrounds that may be used to one or the other.)

We agree and changed the wording to "training" here.

In addition, we added a sentence at the beginning of section 2.3 to explain that we use "training" for machine learning models and "calibration" for the conceptual models as it is typical in the corresponding literature, but that both terms refer to the optimization of a loss function.

Figure 2: Utilize symbols for black and white version distinguishability of basin classification.

(This is now Figure 3) We changed to Okabe-Ito colors (R palette "Okabe-Ito") that are robust under color vision deficiencies and to different symbols for the different altitude classes.

Figure 3/4/5: Particularly the top 2 panels are difficult to read in black and white, double check with the editor whether this would need to be adjusted.

We changed to Okabe-Ito colors (R palette "Okabe-Ito") that are robust under color vision deficiencies.

Line 323-324: Perhaps use a different loss function with guaranteed convexity to avoid this problem. You can still use NSE for evaluation of course to stay consistent with the other models. (I would recommend you do this at least for the reanalysis)

We agree that experimenting with different loss functions is important but this is not a major focus of this paper as we wanted to conform with common practice with LSTM modelling in hydrology, which so far was mainly based on NSE as a loss function.

We also do not see that we could avoid the existence of multiple (many!) local minima of an LSTM model by changing the loss function as the existence of multiple minima originates from the complex model structure and not from the loss function applied to the residuals between model and data. Using different seeds at least partly demonstrates the variability that we can expect from ending up in different local minima.

Line 363: Instead of "was not investigated" saying "this was not the main aim" would be better.

Done.

Line 412-414: Make these claims and future research directions more specific. Why ML and which research will be necessary to achieve this?

We added a statement here that adding catchment properties, such as vegetation and soil, to conceptual or physical hydrological models, could in principle improve the predictive properties of these models under strong input changes. However, the parameterization of the processes for vegetation and soil structures is very difficult. We also added a sentence that machine learning models need to be trained on large variety of catchments with different catchment properties and/or be constrained or preconditioned by physical or biological considerations. This is also further discussed in the final section of the conclusions.

Last paragraph: In the conclusion new ideas should not be first mentioned as it is done here. Citations should also not be in the conclusion section for this reason. Please mention these points earlier in the discussion and just summarize them in the conclusion.

We moved the discussion of hybrid approaches with the citations to a final subsection of the section "Results and Discussion", added some additional citations (those mentioned by CC1 and in our reply to this comment), and mention this point in a shortened final section of the conclusions without citations.

Thank you for your submission! I am looking forward to reviewing the revised version of this manuscript!

**RC1:**

Reichert and co-authors describe a study in which they test the qualitative response (metamorphic testing) of two conceptual hydrologic models and a deep learning model trained on CAMELS-US dataset to perturbed temperature and precipitation, aimed at mimicking the qualitative performance of these models under climate change scenarios. The deep learning model (LSTM) outperforms the conceptual models during calibration and validation, but exhibited unexpected hydrologic response in low-elevation basins when temperature was perturbed. Solely training on the low-elevation basins from CAMELS-US improved this qualitative response to the perturbed temperature, suggesting that fine-tuning or limiting datasets to prediction task may help improve out-of-bounds predictions. I provide some comments below that I think will improve the manuscript.

- Generally, I encourage the authors to attempt to summarize all the basins used for metamorphic testing rather than providing individual plots across all basins in the supplementary. There does not appear to be a reason why certain basins are displayed in the main text vs. supplementary. Summarizing across all basins used will help the reader understand better if the pattern is common without having to look through dozens of individual plots in supplementary.

    We agree that a numerical summary across the basins improves the manuscript and we added the following figure with the summary statistics. As we cannot compare with the truth or with data, we compare the sensitivities of the different approaches by calculating the root mean squared differences of all pairs of approaches and average them over the seeds (if available) and over the catchments within the same class. We included the results for the most important case, the temperature sensitivities of the low altitude catchments in the paper and added the other results to the Supporting Information.

[Figure]

In this figure (the calculation of the numerical values is explained in the text), we clearly see that the LSTM sensitivities approach those of GR4neige (decreasing differences [in mm/d]) when moving from the basic LSTM to the LSTM_211 (first column) and they also approach those of HBV (second column). In parallel to that, we see that the sensitivities of the LSTM deviate more and more from the basic LSTM when moving from the basic LSTM to the LSTM_211. These are the changes discussed in the paper.

Still, we believe that showing one example for each class of catchments in the paper is very important to demonstrate the shapes of the sensitivities and, for being transparent, showing the

results for all chosen catchments in the supporting information is useful to demonstrate that the response is typical for the class. For this reason, we suggest to add the summary table/figure but to keep the examples.

Line 40 and elsewhere: Instead of 'modified driving forces', could you use something more generic like 'out-of-bounds predictions'?

We think that "modified driving forces" more clearly expresses what we do than "out-of-bound predictions".

- Line 114-115: the initial clause seems a bit clunky. Would it be clearer to either remove the first clause entirely or consider placing it as a separate sentence?

We agree and restructured this paragraph.

- Lines 120-139: this is a flat 10% increase in precipitation for every precipitation data point in the dataset? What if there is no rain on a given day? I assume that will still be 0 precipitation increase scenario given the equation 1 but clarification would be helpful.

This is correct, $1.1 \times 0 = 0$. We added a sentence to clarify this and also to motivate the choice of doing a relative rather than an absolute input modification.

- Line 160: can you give examples of what these precipitation or temperature related attributes would be?

The full and reduced sets of attributes are given in Table B1. We added this reference in line 160 and provided "mean daily precipitation" and "fraction of precipitation falling as snow" as examples.

- Line 168: what was the validation NSE range for these catchments?

All individual NSE values for both, calibration and validation periods, are given in the Figures in the Supporting Information. The range for calibration is 0.82-0.92, the range for validation is 0.67-0.91. We added this information also to the paper.

- Line 213: I encourage the authors to include a link to the working repository at the moment, or a draft code release.

The link for the working repository for the conceptual models is https://gitlab.com/p.reichert/hyperflex.
The LSTM code used in this work can be accessed at http://doi.org/10.5281/zenodo.3993880. We added this information to the paper.

- Figure 2: describe in the legend what the numbers next to the points represent.

These are the identifiers of the CAMELS basin. We added this statement to the legend.

- Line 253: clarify that the sensitivities are in relation to the outlet discharge and not the overall model performance – 'sensitivities of the models are essentially negative' makes it sound like the models had a poor/unexpected outcome, but this is quite the opposite. I suggest changing to something like, "The predicted outlet discharge for the GR4neige and HBV models was lower with increased temperature, which is expected … "

$\Delta Q_P$ and $\Delta Q_T$ are defined by the equations (1) and (2), respectively. They are negative if the predicted output is lower than for the base simulation and positive if it is higher. We added an explanation about the meaning of these negative sensitivities.

- Figure 3: Please change the colors of the third panel to be different than the colors used to indicate the different types of models in the other panels? This is confusing to switch the meaning of the colors in the same figure.

  We agree. We removed the colors of the temperature panel and we changed the colors of the other panels to Okabe-Ito colors (R palette "Okabe-Ito") that are robust under color vision deficiencies.

- Figure 3: for the top panel, why is there a y-axis that extends to -4 when there are no negative values? Also seems like the max differences are cutoff at the top of the y-axis, please extend higher.

  We think that it makes sense to have the same scale for the sensitivities across all catchments to make it easier to compare them. The chosen scale if a compromise of seeing sufficient detail and not cutting some of the sensitivities too much.

- Figure 3: for the fourth panel, indicate what the black circles are – I assume they are observations

  Yes, they are. We added this to the legend.

- Figure 4: please include the full legend here so the reader doesn't have to refer to a separate figure

  Done.

- Line 276: change "rainfall" to "precipitation" as a lot of this precipitation is falling as snow in this basin and other basins.

  We agree. Thank you for the hint. We changed this consistently throughout the paper.

- Figure 4: it is hard to distinguish the different model traces on here and I cannot tell what I'm supposed to take away from the bottom panel. I suggest splitting out the LSTM traces into a separate panel and/or show the deviation of the LSTM_x compared to the base LSTM results.

  We changed the colors to Okabe-Ito colors (R palette "Okabe-Ito") that are robust under color vision deficiencies and we modified the text to better explain the key information to be gained from this figure. We point the attention of the readers to the new Figure 8 that provides complementary quantitative information and demonstrates the approach of the LSTM_x to the conceptual models when moving from the original model to the one with the training set reduced to 211 low-level basins.

- Lines 395-402: I'm curious if the authors tried pre-training on the entire dataset with early stopping criteria as to not overfit, and then fine tune on the reduced dataset with only low-elevation basins. That seems like it might be the best of both worlds – providing better fits by using more data but also passing the metamorphic test.

  We agree that strategies like pre-training can be highly effective. In a separate study (Ma et al. 2021, cited in the paper), we have leveraged pre-training and transfer learning technique to enhance the performance on smaller datasets. However, for the purpose of this study, our focus is on employing a standard LSTM model. This approach is intended to provide insights into the typical methodologies used in training ML models and to highlight the discernible differences that emerge from such conventional training practices.

- Lines 398-399: by how much did the quality of fit deteriorate? By 1%, 30%? I suggest adding in a quantitative measure so the readers can evaluate how much the tradeoff is for passing the metamorphic test with less data.

  We agree. Taking the basin 11468500 as an example, when the number of data reduces from 211 to 99, the validation NSE drops from 0.86 to 0.30 which is far below the range of validation NSE values for all approaches used in the paper (0.67-0.91).

- Lines 412-414: See Topp et al. 2023 https://doi.org/10.1029/2022WR033880 for comparison of ML architectures to prediction in unseen conditions. They suggest ML process/physics guidance helps improve predictions in unseen conditions. Likewise, see Read et al. 2019 https://doi.org/10.1029/2019WR024922 for out-of-bounds predictions using process-guidance for ML models.

  Thank the reviewer for these hints. These papers are about stream and lake water temperature which is not a topic of this paper. However, we agree with the reviewer that hybrid modelling approaches that combine mechanistic with machine learning submodels are of great interest in many fields of applied research. However, we cannot provide a literature review about many or all of this fields.

  For this reason, we limit the discussion section to references about modelling catchment outflow and we updated the literature review about such approaches.

---

## Author Response (AR2)

**Referee #1:**

accepted as is

We thank the referee for the positive valuation.

**Referee #2:**

- Line 430: "this this".

  Thank you. Done.

- The new figure describing expected responses is excellent and improves the section greatly.

  Thank you for the suggestion to include such a figure and for the positive feedback.

- The responses to my other previous comments are also satisfactory.

  Thank you for the positive reply.

- The conclusion and summary section is 2 pages long, if possible, try to narrow this down to the key points to be more concise, as there is a lot of information that is more reminiscent of a discussion section rather than a conclusion. I believe aiming for about 1 page total would be good-

  We agree. We moved the discussion of the questions raised in the introduction to the discussion section 3.4 and slightly adapted the conclusions section 4 (which is now less than 1 page).

In addition, we made a few minor changes to the manuscript (see tracked version)

**Editorial assistant:**

Checking your paper, I noticed that your table 2 contains coloured cells. Please note that this will not be possible in the final revised version of the paper due to HTML conversion of the paper. When revising the final version, you can use footnotes or italic/bold font. For now, the process will continue, but please note that the final version cannot be published by using coloured tables.

We reformatted the red numbers into black, bold numbers.

---

## Author Response (AR3)

Dear Editor

Please find the uploaded files. The only changes are those required in formatting the Supporting Information.

Best regards        Peter Reichert